statistics

multinomial counts, exact probability, urns

**Author for correspondence:**
Pasquale Cirillo
e-mail: p.cirillo@tudelft.nl

# Computing the exact distributions of some functions of the ordered multinomial counts: maximum, minimum, range and sums of order statistics

## Marco Bonetti[1], Pasquale Cirillo[2] and Anton Ogay[2]

[1]Dondena Research Center and Bocconi Institute for Data Science and Analytics, Bocconi University, Via G. Roentgen 1, 20136 Milan (MI), Italy
[2]Applied Probability Group, Delft University of Technology, Van Mourik Broekmanweg 6, 2628 Delft, The Netherlands

 PC, 0000-0002-7711-049X

Starting from seminal neglected work by Rappeport (Rappeport 1968 Algorithms and computational procedures for the application of order statistics to queuing problems. PhD thesis, New York University), we revisit and expand on the exact algorithms to compute the distribution of the maximum, the minimum, the range and the sum of the $J$ largest order statistics of a multinomial random vector under the hypothesis of equiprobability. Our exact results can be useful in all those situations in which the multinomial distribution plays an important role, from goodness-of-fit tests to the study of Poisson processes, with applications spanning from biostatistics to finance. We describe the algorithms, motivate their use in statistical testing and illustrate two applications. We also provide the codes and ready-to-use tables of critical values.

## 1. Introduction

The multinomial random vector arises naturally in several statistical problems, from queuing theory to software reliability models, from clinical trials to financial mathematics, from goodness-of-fit tests to transportation problems [1,2]. The multinomial experiment is a common way of representing the multinomial random vector as the result of throwing $n$ independent balls into $m$ independent urns, each with a given probability of attraction $p_i$, and of counting the number $n_i$ of

balls that fall into urn $i$, for $i = 1, \ldots, m$ [3]. The probability mass function (pmf) of the resulting vector of counts $(N_1, \ldots, N_m)^T$ is

$$P(N_1 = n_1, \ldots, N_m = n_m; \mathbf{p}, m, n) = \frac{n!}{n_1! \cdots n_m!} \prod_{i=1}^{m} p_i^{n_i},$$

where $\mathbf{p} = (p_1, \ldots, p_m)^T$ and $\sum_{k=1}^{m} n_k = n$.

A common statistical hypothesis of interest is whether the underlying multinomial distribution is equiprobable, so that the probability for a ball of falling in any of the urns is always the same, i.e. $H_0 : P = P_0$ versus $H_1 : P \neq P_0$, where $P_0$ is the equiprobable pmf

$$P_0(N_1 = n_1, \ldots, N_m = n_m; \mathbf{p} = (p, \ldots, p)^T, m, n) = \frac{n!}{n_1! n_2! \ldots n_m!} p^n, \tag{1.1}$$

with $p = 1/m$. From now on, we indicate the pmf in equation (1.1) as $\text{Mult}(n, \mathbf{p_0})$, where $\mathbf{p_0} = (m^{-1}, \ldots, m^{-1})^T$.

Many procedures have been proposed to test the equiprobability hypothesis, a good review being [2]. The classical way is to use the $\chi^2$ goodness-of-fit test, first introduced by Pearson [4] and based on the statistic

$$X^2 = \sum_{i=1}^{m} \frac{(N_i - np)^2}{np}.$$

Other approaches replace $X^2$ with the Neyman modified $X^2$ [5], with the log-likelihood ratio statistic

$$G^2 = 2 \sum_{i=1}^{m} N_i \ln\left(\frac{N_i}{np}\right),$$

or with the Freeman–Tukey statistic [6].

In 1962, Young [7] revisited this problem and proposed two alternatives based on the rescaled range

$$W_m = \left( \max_{1 \leq k \leq m} \frac{N_k}{n} - \min_{1 \leq l \leq m} \frac{N_l}{n} \right) \tag{1.2}$$

and the rescaled mean

$$M_m = (mn)^{-1/2} \sum_{i=1}^{m} \left| N_i - \frac{n}{m} \right|.$$

These new statistics revealed some power advantages under certain alternatives [7].

In testing the equiprobability hypothesis, all the statistics above rely on approximations (like the Normal, the $\chi^2$, the Beta, the Dirichlet or the Gumbel), being their exact distributions not known. This requires that the original data satisfy some often heuristic conditions: for instance, the $\chi^2$ approximation for the Pearson statistic is typically recommended when $n \geq 5\,m$ [2,6]. In several applications, especially when dealing with small samples, these conditions are rarely satisfied and as a consequence the tests may be unreliable.

More recently, Corrado [8] has offered a solution, based on what he calls stochastic matrices, for computing the exact probabilities of the multinomial maximum, minimum and range. Corrado's approach clearly solves the problem of using potentially inaccurate approximations but—as we shall see—it represents an unsatisfactory solution, since it requires ad hoc computations for each combination of $n$ and $m$. Our aim is thus to propose a more general and flexible approach.

Our investigation originates from an old PhD thesis by Rappeport [9], in which two algorithms for the exact computation of the distributions of the multinomial maximum and of the sum of the three largest multinomial-order statistics were proposed. We first describe Rappeport's results, and then we present novel general algorithms for computing the exact distributions of the multinomial minimum, of the range and of the sum of the $J$ largest order statistics. This means that, for example, the distribution of the test statistic $W_n$ in equation (1.2) can now be obtained exactly.

The article develops as follows. Section 2 provides a quick overview of the existing distributional approximations for the multinomial range and other order statistics, whereas §3 contains the exact results of Corrado [8]. Section 4 is devoted to the original algorithms by Rappeport, while §§5 and 6 introduce our results for the minimum and the range, respectively. Section 7 discusses some non-trivial accuracy issues of the commonly used approximations. In §8, we provide some additional motivation for the use of the (sums of) the largest multinomial counts in hypothesis testing. In §9, we shortly describe some of the many possible applications that involve these exact results. We close with

a discussion in §10. Appendix A contains tables of critical values for the multinomial maximum, minimum, range and sums, as well as codes for all algorithms.

# 2. Approximations

Starting from some results of Pearson & Hartley [10], a first approximation for the multinomial distribution under equiprobability was introduced by Johnson & Young [11]. Young [7] used this approximation to derive a limiting distribution for the range of the multinomial sample.

The limiting distribution of the maximum, using a Gumbel approximation, was initially introduced by Kolchin *et al.* in 1978 [12], with some errors that were later corrected in Dasgupta [13]. No general result appears to be available for the multinomial minimum or other order statistics. Some specific cases, easily computable by hand, are described in [2].

Note that, given the increased (and still increasing) computing power one can rely upon today, the probability distributions of functions of the multinomial counts can also be estimated via Monte Carlo simulations. However, even if extremely accurate, from a conceptual point of view they are still approximations and not exact results, as those we will discuss in this article.

All the approximations presented below are derived under the hypothesis of equiprobability.

## 2.1. Approximation of the distribution of the range

It is well known that, marginally, for $i = 1, \ldots, m$ one has $N_i \sim \text{Binom}(n, p_i)$, so that under the null hypothesis $E(N_i) = n\, p_i = n\, p$ and $\text{Var}(N_i) = n\, p\, (1-p)$ with $p = 1/m$. Using the multidimensional central limit theorem, the joint distribution of the standardized multinomial vector $(\omega_1, \ldots, \omega_m)^{\text{T}}$, with

$$\omega_i = \frac{N_i - np}{\sqrt{np(1-p)}} = \frac{mN_i - n}{\sqrt{n(m-1)}},$$

converges in distribution, as $n \to \infty$, to a multivariate normal distribution with zero mean vector, unit variances and covariance between $\omega_i$ and $\omega_j$ equal to $(1-m)^{-1}$ for $i \neq j$. Note that the limiting distribution is actually degenerate, i.e. its support is $(m-1)$-dimensional due to the $n$-sum constraint that applies to the $N_i$ terms (or, equivalently, the zero-sum constraint on the $\omega_i$ terms). One has a non-degenerate limiting distribution for any choice of a set of $m-1$ terms from the set $(\omega_1, \ldots, \omega_m)$.

The distribution of the range of $m$ identically independently distributed (i.i.d.) standard normal variables $X_1, X_2, \ldots, X_m$ is a known quantity, and it can be computed as

$$P\left( \max_{1 \le i \le m} X_i - \min_{1 \le j \le m} X_j \le r \right) = m \int_{-\infty}^{\infty} \phi(x) \left( \int_{x}^{x+r} \phi(u)\, \mathrm{d}u \right)^{m-1} \mathrm{d}x,$$

where $\phi(x)$ is the probability density function of a standard normal random variable. Using this, Young [7] shows that the distribution of the scaled multinomial range can be approximated as

$$P\left( \max_{1 \le i \le m} \frac{N_i}{n} - \min_{1 \le j \le m} \frac{N_j}{n} \le r \right) \approx P\left( \max_{1 \le i \le m} X_i - \min_{1 \le j \le m} X_j \le (r + \delta_m)\sqrt{nm} \right),$$

where $\delta_m$ is a continuity factor such that

$$\delta_m = \frac{1}{n} \quad \text{for } m = 2 \quad \text{and} \quad \delta_m = \frac{1}{2n} \quad \text{for } m > 2.$$

The approximation works best for large values of the ratio $n/m$ (our simulations suggest $n > 5m$). We refer to [2,7] for additional details.

## 2.2. Approximation of the distribution of the maximum

The approximating distribution for the maximum of a multinomial sample was proposed by Kolchin *et al.* [12] and improved by Dasgupta [13], to which we refer for all technical details.

Set

$$\mu = \frac{n}{m} \quad \text{and} \quad \kappa = \frac{\log m - \frac{1}{2} \log \log m}{\mu},$$

and let $\epsilon$ be the unique positive root of the equation

$$(1 + \epsilon) \log(1 + \epsilon) - \epsilon = \kappa.$$

Then the law of the maximum of a multinomial sample converges in distribution to a Gumbel random variable, i.e.

$$P\left( \frac{\max_{1 \leq i \leq m} N_i - \mu(1 + \epsilon)}{\sqrt{n/2m \log m}} + \frac{\log(4\pi)}{2} \leq z \right) \xrightarrow{d} e^{-e^{-z}},$$

for all real $z$, as $n \to \infty$. As observed in [2,8], the approximation by Kolchin and Dasgupta is the best one available for the multinomial maximum in the literature so far.

# 3. Exact results by Corrado (2011)

Corrado's approach [8] is based on a matrix representation for the construction of the transition probabilities for the number of balls in the different urns. The main advantage of Corrado's method is that it does not require equiprobability.

Let $N_k$ be the random number of balls in urn $k$. The sequence $S_k = S_{k-1} + N_k$ describes the cumulative ball count from $S_0 = 0$ to $S_m = n$, where $m$ is the total number of urns. The transition probability from $S_{k-1}$ to $S_k$, i.e. $P(S_k = s_k \mid S_{k-1} = s_{k-1}; p^*_k)$, for brevity $P(s_k \mid s_{k-1}; p^*_k)$, is equal to

$$P(s_k | s_{k-1}; p^*_k) = \begin{cases} \binom{n - s_k}{s_k - s_{k-1}} (p^*_k)^{s_k - s_{k-1}} (1 - p^*_k)^{n - s_k} & s_k \geq s_k - 1 \\ 0 & \text{otherwise} \end{cases}, \tag{3.1}$$

where $p^*_k = p_k / \sum_{j=k}^{m} p_j$ (so that in particular $p^*_1 = p_1$).

From equation (3.1), for $k = 1, \dots, m$, we can determine upper-triangular matrices of the form

$$Q_k = \begin{bmatrix} P(0 | 0; p^*_k) & P(1 | 0; p^*_k) & \dots & P(n | 0; p^*_k) \\ 0 & P(1 | 1; p^*_k) & \dots & P(n | 1; p^*_k) \\ \dots & \dots & \dots & \dots \\ 0 & 0 & \dots & 1 \end{bmatrix}.$$

These matrices provide a straightforward way to calculate the desired exact probabilities of the multinomial order statistics. The first transition from $s_0 = 0$ to $s_1$ travels across a starting row vector $Q_1^1$ defined as

$$Q_1^1 = [P(0 | 0; p_1) \quad P(1 | 0; p_1) \quad \dots \quad P(n | 0; p_1)].$$

$Q_1^1$ indicates the first row of matrix $Q_1$. The product $Q_1^1 \times Q_2$ is a row vector whose elements give the convolution distribution of $S_2 = N_1 + N_2$, while $Q_1^1 \times Q_2 \times \cdots Q_k$ represents the convolution distribution of the random sum $S_k$. Naturally $Q_1^1 \times Q_2 \times \cdots Q_m = 1$ concentrates all the mass on $S_m = n$, so that $Q_m$ is equal to a column vector of ones.

In order to calculate the exact probability for the maximum amount of balls in the urns not to exceed a given $r$, i.e. $P(\max_{1 \leq i \leq m} N_i \leq r)$, Corrado suggests the following procedure:

(i) In all the matrices $Q_k$, $k = 1, \dots m$, set to 0 all the transition probabilities $P(s_k | s_{k-1}; p^*_k)$ for which $s_k - s_{k-1} > r$. This defines a new series of sub-matrices, $Q_1^*, \dots, Q_m^*$, called *culled*.
(ii) The product of the sub-matrices $Q_1^*, \dots, Q_m^*$ gives the exact probability of $P(\max_{1 \leq i \leq m} N_i \leq r)$.

A simple example taken from [8] will clarify the method. Imagine to throw three balls across three urns, so that $n = m = 3$. Then we can easily verify that $Q_1^1 = [0.296 \ 0.444 \ 0.222 \ 0.037]$, $Q_3 = [1 \ 1 \ 1 \ 1]$ and

$$Q_2 = \begin{bmatrix} 0.125 & 0.375 & 0.375 & 0.125 \\ 0 & 0.25 & 0.5 & 0.25 \\ 0 & 0 & 0.5 & 0.25 \\ 0 & 0 & 0.5 & 0.5 \\ 0 & 0 & 0 & 1 \end{bmatrix}.$$

Now, suppose that one is interested in computing the probability that the multinomial maximum is smaller than or equal to 2. The matrices above need to be modified as $Q_1^{*1} = [0.296 \ 0.444 \ 0.222 \ 0]$, $Q_3^* = [0 \ 1 \ 1 \ 1]$ and

$$Q_2^* = \begin{bmatrix} 0.125 & 0.375 & 0.375 & 0 \\ 0 & 0.25 & 0.5 & 0.25 \\ 0 & 0 & 0.5 & 0.25 \\ 0 & 0 & 0.5 & 0.5 \\ 0 & 0 & 0 & 1 \end{bmatrix}.$$

Then $P(\max_{1 \leq i \leq m} N_i \leq 2) = Q_1^{*1} \times Q_2^* \times Q_3^* = 0.889$.

The distribution of the minimum can be obtained similarly, if one modifies each matrix $Q_k$, by setting $P(s_k \,|\, s_{k-1}; p_k^*)$ equal to 0 for $s_k - s_{k-1} < r$ (notice the change in the inequality sign). Additional details are available in [8].

Interestingly, the distribution of the multinomial range can also be computed using the matrix representation. Set $Q_k(a_k, b_k)$ to be the culled matrix for urn $k$, where $P(s_k \,|\, s_{k-1}; p_k^*) = 0$ for all $N_k > a_k$ or $N_k < b_k$. Introducing the set of all possible allocations of $n$ balls across $m$ urns as $\bigcap_{k=1}^m \{a_k \geq N_k \geq b_k\}$, one can express the joint probability of the maximum and the minimum ball counts as

$$P\left(\bigcap_{k=1}^m a_k \geq N_k \geq b_k\right) = Q_1^{*1} \times \prod_{k=2}^{m-1} Q_k(a_k, b_k) \times Q_m^*. \tag{3.2}$$

Note that the set of allocations described above have intersecting intervals, so that to compute the exact probabilities for the range the intersection probabilities should be subtracted:

$$P\left(\max_{1 \leq k \leq m} N_k - \min_{1 \leq l \leq m} N_l < r\right) = \sum_{h=0}^{n-r+1} Q_1^{*1} \times \prod_{k=2}^m Q_k(h + r - 1, h) \times Q_m^*$$
$$- \sum_{h=0}^{n-r} Q_1^{*1} \times \prod_{k=1}^m Q_k(h + r - 1, h + 1) \times Q_m^*.$$

Corrado's method works nicely and allows for the exact probability computations of the multinomial maximum, minimum and range. However, it has a strong limitation: for every new composition, and for every value of $r$, the culled matrices have to be redesigned and recalculated, something not very efficient.

# 4. Rappeport's algorithms

In 1968, Rappeport [9] proposed two iterative algorithms for the distribution of the multinomial maximum, and for the sum of the three largest multinomial order statistics. That work remained unpublished, and Rapperport's idea has been mostly ignored. These algorithms are based on the representation of all the possible outcomes of the multinomial experiment in the form of a tree. The desired probabilities are computed by moving across the branches of this tree according to certain rules. A relevant feature of Rappeport's approach is the possibility of deriving a general algorithm, which does not require adjustments that depend on the specific parameters and values of interest.

Consider six balls thrown across three equiprobable urns. The possible outcomes of this multinomial experiment can be represented with the tree in figure 1, where each branch corresponds to a particular partition (up to rearrangement) of the balls into the urns. For instance, the blue path on the left, characterized by the squares with bold edges, represents the situation in which one of the urns contains all the six balls, while the other two are empty. Conversely, the green path with the dotted squares represents the configuration where one urn contains three balls, one urn contains two balls and the remaining urn contains only one ball. And so on for all the other branches.

Rappeport also discusses the case of urns with different attraction probabilities, under the assumption that they can be collected into two or three same-probability groups. In these non-equiprobable situations, Rappeport's approach loses its generality, and it necessarily requires case-specific adjustments that do depend on the characteristics of the groups of urns. For this reason, and since the main null hypothesis in multinomial statistical tests is that of equiprobability, we do not discuss the details of the two- and three-group cases.

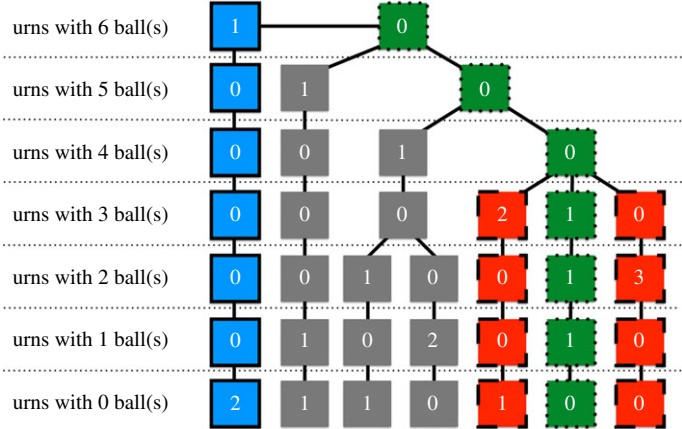

**Figure 1.** Tree representation of the possible outcomes of the multinomial experiment of throwing six balls into three equiprobable urns.

## 4.1. Distribution of the maximum

Let $N_{\langle 1 \rangle}, \ldots, N_{\langle m \rangle}$ be the order statistics of the multinomial counts $N_1, \ldots, N_m$ in descending order, so that $N_{\langle 1 \rangle}$ is the maximum, $N_{\langle 2 \rangle}$ is the second largest order statistics, and so on, up to the minimum $N_{\langle m \rangle}$. Note that, just like the counts, the ordered counts clearly need not be all different.

In the tree representation, to compute the exact probabilities $P(N_{\langle 1 \rangle} \le r; n, m)$ of the multinomial maximum under equiprobability, one sums the probabilities of all the paths, whose nodes are characterized by zeros for all the levels from $r+1$ to $n$. For example, suppose that one wants to compute the probability that no urn contains more than $r = 3$ balls, when $n = 6$ and $m = 3$. Then she needs to sum the probabilities of the green dotted and the red dashed paths in figure 1, because in all the other situations (grey and blue) $N_{\langle 1 \rangle} \ge 4$.

To perform such computations, Rappeport develops an efficient iterative procedure, in which the probability $P(N_{\langle 1 \rangle} \le r; n, m)$ is explicitly obtained from $P(N_{\langle 1 \rangle} \le r-1; n, m)$ and some initial value.

Let us compute the probability that the maximum number of balls in the $m$ urns is exactly equal to $r$, and let us assume that such a maximum is unique, i.e. all the others urns have at most $r-1$ balls. Using equation (1.1), and introducing the operator $W_2$, which is the sum over all possible values $n_{\langle 2 \rangle}, \ldots, n_{\langle m \rangle}$ such that $n_{\langle 1 \rangle} > n_{\langle 2 \rangle}$, we can write

$$P(N_{\langle 1 \rangle} = r \mid N_{\langle 1 \rangle} > N_{\langle 2 \rangle}; n, m) = \frac{1}{r!} W_2 \left( \frac{n!}{m^n \prod_{i=2}^{m} n_{\langle i \rangle}!} \frac{m!}{\prod_{k=0}^{r-1} ((\#n_i = k)!)} \right).$$

Forcing notation, here $(\#n_i = k)$ denotes the number of $n_i$s equal to $k$, that is to say the number of urns containing $k$ balls. The fraction $m! / \prod_{k=0}^{r-1} (\#n_i = k)!$ arises from a simple combinatorial argument: since the urns are equiprobable, the probability of a given composition of $n_1, \ldots, n_m$ should be multiplied by the total number of its unique permutations.

Let us now relax the assumption of uniqueness for the maximum, to obtain

$$P(N_{\langle 1 \rangle} = r \mid N_{\langle 1 \rangle} = N_{\langle q \rangle} > N_{\langle q+1 \rangle}; n, m)$$

$$= \frac{1}{(r!)^q q!} W_{q+1} \left( \frac{n!}{m^n \prod_{i=q+1}^{m} n_{\langle i \rangle}!} \frac{m!}{\prod_{k=0}^{r-1} ((\#n_i = k)!)} \right).$$

Let $q = 0$ indicate the case of $n_{\langle 1 \rangle} < r$. Summing over $q$ then yields

$$P(N_{\langle 1 \rangle} \le r; n, m) = \sum_q \frac{1}{(r!)^q q!} \frac{n! m!}{m^n}$$

$$\times W_{q+1} \left( \frac{1}{\prod_{i=q+1}^{m} (n_{\langle i \rangle}!) \prod_{k=0}^{r-1} ((\#n_i = k)!)} \right).$$

(4.1)

The number of urns containing exactly $r$ balls cannot be greater than $\lfloor n/r \rfloor$, which therefore defines the upper limit of the summation. Moreover, since all the other $n - rq$ balls should be placed in the

remaining $m - q$ urns, with maximum not exceeding $r - 1$, one must have $(m - q)(r - 1) \geq (n - rq)$, therefore the lower limit for $q$ is max $(0, n - rm + m)$. The range of the summation in equation (4.1) thus becomes

$$\max(0, n - rm + m) \leq q \leq \left\lfloor \frac{n}{r} \right\rfloor. \tag{4.2}$$

Noting that

$$W_{q+1}\left( \frac{1}{\prod_{i=q+1}^{m}(n_{\langle i \rangle}!) \prod_{k=0}^{r-1}((\#n_i = k)!)} \frac{(m - q)!(n - rq)!}{(m - q)^{n - rq}} \right) = P(N_{\langle 1 \rangle} \leq r - 1; n - rq, m - q),$$

the following iterative formula for the probability of the multinomial maximum holds:

$$P(N_{\langle 1 \rangle} \leq r; n, m) = \sum_q A_q P(N_{\langle 1 \rangle} \leq r - 1; n - rq, m - q), \tag{4.3}$$

where

$$A_q = \frac{n!m!}{m^n} \frac{1}{(r!)^q q!} \frac{(m - q)^{n - rq}}{(m - q)!(n - rq)!}.$$

The starting point of the iteration is represented by the probability that the maximum is smaller than or equal to 1 (where the former clearly cannot occur for $n, m > 0$), i.e.

$$P(N_{\langle 1 \rangle} \leq 1; n, m) = P(N_{\langle 1 \rangle} = 1; n, m) = \begin{cases} \frac{m!}{m^n (m - n)!} & \text{if } m \geq n \\ 0 & \text{if } m < n \end{cases}.$$

This quantity can be easily derived from equation (1.1), since the only possible configurations corresponding to $\{N_{\langle 1 \rangle} = 1\}$ are those with $n$ frequencies equal to 1 and $n - m$ frequencies equal to 0, and there are $\binom{m}{n}$ such sequences.

Table 3 in appendix A contains critical values for the maximum as obtained with this algorithm, for different combinations of $n$ and $m$. As expected, these exact numbers coincide with those obtainable using Corrado's stochastic matrices [8] (and the same will be true for the minimum and the range).

## 4.2. Distribution of the sum of the $J$ largest order statistics

The algorithm for the maximum can also be used for the calculation of the exact distribution of the sum of the $J$ largest order statistics. Rappeport discusses explicitly the cases $J = 2, 3$, and only gives some hints about the general case $3 < J < m$.

Consider the case $J = 3$ as in the original work by Rappeport.[1] To compute $P(N_{\langle 1 \rangle} + N_{\langle 2 \rangle} + N_{\langle 3 \rangle} \leq r; n, m)$, one may partition this probability into different terms, corresponding to the different possible ranges of $N_{\langle 1 \rangle}$ and $N_{\langle 2 \rangle}$. Indeed, one can distinguish among three disjoint cases:

(i) $N_{\langle 1 \rangle} \leq r/3$. In this case,

$$P\left( \sum_{i=1}^{3} N_{\langle i \rangle} \leq r; n, m \right) = P\left( N_{\langle 1 \rangle} \leq \frac{r}{3}; n, m \right).$$

(ii) $N_{\langle 1 \rangle} > r/3$ and $N_{\langle 2 \rangle} \leq (r - N_{\langle 1 \rangle})/2$. Here one can fix a value of $N_{\langle 1 \rangle} = t_1$, thus forcing one urn to contain exactly $t_1$ balls. The remaining urns define a smaller sample with $n^* = n - t_1$ and $m^* = m - 1$. If the maximum of this new sample is smaller or equal than $(r - t_1)/2$, the original inequality for the sum of the three largest order statistics will automatically hold. The total probability in this case is equal to the sum over all possible values of $t_1$, i.e.

$$P\left( \sum_{i=1}^{3} N_{\langle i \rangle} \leq r; n, m \right) = \sum_{t_1 = \lfloor r/3 + 1 \rfloor}^{r} A_{t_1} P\left( N_{\langle 1 \rangle} \leq \frac{r - t_1}{2}; n - t_1, m - 1 \right),$$

[1] The original formulation of Rappeport contained a few errors, here corrected.

with

$$A_{t_1} = \frac{n!m!}{m^n} \frac{1}{t_1!} \frac{(m-1)^{n-t_1}}{(m-1)!(n-t_1)!}.$$

(iii) $N_{\langle 1 \rangle} > r/3$ and $N_{\langle 2 \rangle} > (r - N_{\langle 1 \rangle})/2$. One may proceed as in the previous case, but now both values of $N_{\langle 1 \rangle}$ and $N_{\langle 2 \rangle}$ must be fixed, so that

$$P\left(\sum_{i=1}^{3} N_{\langle i \rangle} \le r; n, m\right) = \sum_{t_1 = \lfloor r/3+1 \rfloor}^{r-1} \sum_{t_2 = \lfloor (r-t_1)/2+1 \rfloor}^{\min(t_1, r-t_1)} A_{t_1,t_2} B_{t_1,t_2}$$
$$\times P(N_{\langle 1 \rangle} \le r - t_1 - t_2; n - t_1 - t_2, m - 2),$$

where

$$A_{t_1,t_2} = \frac{n!m!}{m^n} \frac{(m-2)^{n-t_1-t_2}}{(m-2)!(n-t_1-t_2)!} \frac{1}{t_1!t_2!},$$

and $B_{t_1,t_2} = 0.5$ if $t_1 = t_2$, and it is equal to 1 otherwise. The term $B_{t_1,t_2}$ accounts for the possibility of $t_1$ and $t_2$ being equal.

Collecting the probabilities from points (i), (ii) and (iii) above yields the desired probability.

The more general distribution of the sum of the $J < m$ largest order statistics can be computed similar to the $J = 3$ case, by splitting the probability $P\left(\sum_{i=1}^{J} N_{\langle i \rangle} \le r; N, m\right)$ into $J$ terms, each one dealing with some combinations of values for the first $J - 1$ order statistics.

The general explicit formula for the distribution of the sum of the $J$ largest order statistics is

$$P\left(\sum_{i=1}^{J} N_{\langle i \rangle} \le r; n, m\right)$$
$$= P\left(N_{\langle 1 \rangle} \le \frac{r}{J}; n, m\right) + \sum_{t_1 = \lfloor r/J+1 \rfloor}^{r} A_{t_1} P\left(N_{\langle 1 \rangle} \le \frac{r - t_1}{J-1}; n - t_1, m - 1\right) + \cdots$$
$$+ \sum_{t_1} \cdots \sum_{t_{J-1}} A_{t_1,\ldots,t_{J-1}} B_{t_1,\ldots,t_{J-1}} \times P\left(N_{\langle 1 \rangle} \le r - \sum_{i=1}^{J-1} t_i; n - \sum_{i=1}^{J-1} t_i, m - J + 1\right). \quad (4.4)$$

If we denote with $I$ the total number of summations for a particular range of values, then the relative summation limits are defined as

$$\begin{cases} \left\lfloor \frac{r}{J} + 1 \right\rfloor \le t_1 \le r - I + 1 & \text{if } i = 1 \\ \left\lfloor \frac{r - \sum_{i=1}^{I-1} t_i}{J - I + 1} + 1 \right\rfloor \le t_i \le \min\left(t_{I-1}, r - \sum_{i=1}^{I-1} t_i\right) & \text{if } 2 \le i \le I \end{cases}.$$

The coefficients $A$ and $B$ are calculated according to the formulae

$$A_{t_1,\ldots,t_I} = \frac{n!m!}{m^n} \frac{1}{\prod_{i=1}^{I} (n_{\langle i \rangle})!} \frac{(m-I)^{(n-\sum_{i=1}^{I} t_i)}}{(m-I)!(n - \sum_{i=1}^{I} t_i)!} \quad (4.5)$$

and

$$B_{t_1,\ldots,t_I} = \frac{1}{\prod_{k=t_1}^{t_I} (\#t_i = k)!},$$

where, again, $(\#t_i = k)$ denotes the number of $t_i$s equal to $k$.

In appendix A, we report Matlab code to compute the distribution of the sum of the $J < m$ largest order statistics. In tables 4 and 5, we provide critical values for $J = 2$ and $J = 3$. Importantly, note that the algorithm for the sum can also be used immediately to compute the exact probabilities of the second, the third and all the other order statistics.

## 5. The exact distribution of the multinomial minimum

The distribution of the smallest order statistic $N_{\langle m \rangle}$ can be easily derived, by using the probability of the sum of the $m - 1$ largest order statistics, given that $P(N_{\langle m \rangle} \ge r; n, m) = P\left(\sum_{i=1}^{m-1} N_{\langle i \rangle} \le n - r; n, m\right)$.

However, it is not difficult to see that this approach turns out to be very computationally inefficient already for quite small values of $n$ and $m$.

A new, efficient algorithm for the multinomial minimum can be constructed starting from that of the multinomial maximum. One needs to slightly modify the way in which we move across the branches of the tree. We start by assigning probability 0 to all the branches of the tree that contain urns with less than $r$ balls. By iterating through all the possible values of the maximum, from $r$ to $n$, we then compute

$$P(N_{\langle m\rangle} \geq r) = \sum_{t=r}^{n-1} P(N_{\langle 1\rangle} \leq t; n, m), \tag{5.1}$$

with the conditions $P(N_{\langle 1\rangle} \leq r - 1; n, m) = 1$ if $n = m = 0$, and $P(N_{\langle 1\rangle} \leq r - 1; n, m) = 0$ if $n \neq 0$ or $m \neq 0$. Then, by computing $1 - P(N_{\langle m\rangle} \geq r + 1)$, one can easily obtain the probability mass function of the multinomial minimum.

The Matlab code is once again available in appendix A, and table 6 provides some useful critical values.

# 6. The exact distribution of the multinomial range

We now introduce a new iterative algorithm for computing the exact distribution of the multinomial range. Unlike the solution proposed by Corrado [8], our approach does not require any modification of the algorithm for every new urn composition.

Back to our earlier example with $n = 6$ balls and $m = 3$ urns, consider the probability that the range is smaller than or equal to 3. Such a probability can be split into two terms corresponding to the different ranges of variation of the maximum. That is,

$$P(N_{\langle 1\rangle} - N_{\langle 3\rangle} \leq 3) = P(N_{\langle 1\rangle} - N_{\langle 3\rangle} \leq 3 \,|\, N_{\langle 1\rangle} \leq 3) + P(N_{\langle 1\rangle} - N_{\langle 3\rangle} \leq 3 \,|\, N_{\langle 1\rangle} > 3).$$

The first term can be computed using equation (4.3), since $P(N_{\langle 1\rangle} - N_{\langle 3\rangle} \leq 3 \,|\, N_{\langle 1\rangle} \leq 3) = P(N_{\langle 1\rangle} \leq 3)$.

To compute the second term, one may use a procedure similar to the one for the minimum: to iterate through all the possible values for the maximum, while assigning zero probability to all the branches that have urns with less balls than the current maximum, i.e. three balls in our example. For instance, we compute $P(N_{\langle 1\rangle} \leq 4 \,|\, N_{\langle 1\rangle} > 3)$ with the additional conditions that $P(N_{\langle 1\rangle} < 1; n, m) = 1$, if $n = m = 0$, and $P(N_{\langle 1\rangle} < 1; n, m) = 0$, when $n \neq 0$ or $m \neq 0$, so to avoid meaningless paths.

The general algorithm is therefore as follows:

$$P(N_{\langle 1\rangle} - N_{\langle m\rangle} \leq r; n, m) = P(N_{\langle 1\rangle} \leq r; n, m) + P(N_{\langle 1\rangle} - N_{\langle m\rangle} \leq r \,|\, N_{\langle 1\rangle} > r; n, m). \tag{6.1}$$

The first term is easily computed using the algorithm for the maximum, so we focus our attention on the second term. Assume, for the time being, that $N_{\langle 1\rangle} = N_{\langle q\rangle} = r + 1 > N_{\langle q+1\rangle}$. Then, one can rewrite the second term on the right-hand side of equation (6.1) as

$$P(N_{\langle 1\rangle} - N_{\langle m\rangle} \leq r | N_{\langle 1\rangle} = N_{\langle q\rangle} = r + 1 > N_{\langle q+1\rangle}; n, m) = \frac{F_{q+1,1}\left(\dfrac{n!\, m!}{m^n \prod_{i=q+1}^{m} (n_{\langle i\rangle}!) \prod_{k=t-r}^{r} ((\#n_i = m)!)}\right)}{((r+1)!)^q\, q!},$$

where $F_{q+1,1}$ is the operator that sums over all possible values of $n_{\langle q+1\rangle}, \ldots, n_{\langle m\rangle}$ such that the minimum $n_{\langle m\rangle} \geq 1$, and $n_{\langle 1\rangle} = n_{\langle q\rangle} > n_{\langle q+1\rangle}$. Summing over $q$ thus gives the total probability

$$P(N_{\langle 1\rangle} - N_{\langle m\rangle} \leq r | N_{\langle 1\rangle} = r + 1; n, m)$$
$$= \sum_{q} \frac{n!\, m!}{m^n((r+1)!)^q q!} \times F_{q+1,1}\left(\frac{1}{\prod_{i=q+1}^{m} (n_{\langle i\rangle}!) \prod_{k=t-r}^{r} ((\#n_i = m)!)}\right). \tag{6.2}$$

In this case, $N_{\langle 1\rangle} > r$, and the term corresponding to $q = 0$ should be excluded, so that

$$\max(1, n - (r + 1)m + m) \leq q \leq \left\lfloor \frac{n}{r+1} \right\rfloor.$$

Now, multiplying and dividing equation (6.2) by $[(m-q)^{(n-(r+1)q)}][(m-q)!\,(Nn(r+1)q)!]^{-1}$ yields

$$P(N_{\langle 1 \rangle} - N_{\langle m \rangle} \le r \mid N_{\langle 1 \rangle} = r+1; n, m)$$

$$= \sum_q \frac{n!m!}{m^n((r+1)!)^q q!} \frac{(m-q)^{(n-(r+1)q)}}{(m-q)!(N-(r+1)q)!}$$

$$\times F_{q+1,1}\left( \frac{(m-q)!(n-(r+1)q)!}{(m-q)^{(n-(r+1)q)}} \frac{1}{\prod_{i=q+1}^m n_{\langle i \rangle}! \prod_{k=t-r}^r (\#n_i = k)!} \right). \tag{6.3}$$

Note that the term

$$F_{q+1,1}\left( \frac{1}{\prod_{i=q+1}^m n_{\langle i \rangle}! \prod_{k=t-r}^r (\#n_i = k)!} \frac{(m-q)!(n-(r+1)q)!}{(m-q)^{(n-(r+1)q)}} \right)$$

is an alternative way of writing

$$P(N_{\langle m \rangle} \ge 1 \mid N_{\langle 1 \rangle} \le r; n-(r+1)q, m-q).$$

The condition $N_{\langle 1 \rangle} \le r$ can be imposed to the algorithm by changing the threshold for the maximum. In other words,

$$P(N_{\langle m \rangle} \ge 1 \mid N_{\langle 1 \rangle} \le r; n-(r+1)q, m-q) = P(N_{\langle 1 \rangle} \le r; n, m),$$

with, similarly to what we have seen before, $P(N_{\langle 1 \rangle} \le 0; n, m) = 1$, when $n = m = 0$, and $P(N_{\langle 1 \rangle} \le 0; n, m) = 0$, if $n \ne 0$ or $m \ne 0$.

Plugging this into equation (6.3) produces

$$P(N_{\langle 1 \rangle} - N_{\langle m \rangle} \le r \mid N_{\langle 1 \rangle} = r+1; n, m)$$

$$= \sum_q \frac{1}{((r+1)!)^q q!} \frac{n!m!}{m^n} = \frac{(m-q)^{(n-(r+1)q)}}{(m-q)!(n-(r+1)q)!} \times P(N_{\langle 1 \rangle} \le r; n-(r+1)q, m-q),$$

with the same conditions on $P(N_{\langle 1 \rangle} \le 0; n, m)$.

Recalling equation (4.3) for the distribution of the maximum, we can finally calculate the second term on the r.h.s. of equation (6.1), by summing over all the possible values of $N_{\langle 1 \rangle}$. Thus

$$P(N_{\langle 1 \rangle} - N_{\langle m \rangle} \le r \mid N_{\langle 1 \rangle} > r; n, m) = \sum_{t=r+1}^n P(N_{\langle 1 \rangle} \le t \mid N_{\langle 1 \rangle} > t-1; n, m),$$

where $P(N_{\langle 1 \rangle} \le t - r - 1; n, m) = 1$, if $n = m = 0$, and $P(N_{\langle 1 \rangle} \le t - r - 1; n, m) = 0$, if $n \ne 0$ or $m \ne 0$.

The condition $N_{\langle 1 \rangle} > t - 1$ is introduced at every summation step in order to avoid multiple calculations for the same branches of the outcome tree. This results in different ranges of summation in the recursion, i.e.

$$\max(1, n - tm + m) \le q \le \left\lfloor \frac{n}{t} \right\rfloor \quad \text{if } N_{\langle 1 \rangle} > r$$

and

$$\max(0, n - tm + m) \le q \le \left\lfloor \frac{n}{t} \right\rfloor \quad \text{if } N_{\langle 1 \rangle} \le r.$$

Summarizing, the distribution of the range can be evaluated via the iteration step

$$P(N_{\langle 1 \rangle} - N_{\langle m \rangle} \le r; n, m) = P(N_{\langle 1 \rangle} \le r; n, m) + \sum_{t=r+1}^n P(N_{\langle 1 \rangle} \le t \mid N_{\langle 1 \rangle} > t-1; n, m), \tag{6.4}$$

with the same conditions as above for $P(N_{\langle 1 \rangle} \le t - r - 1; n, m)$, when $N_{\langle 1 \rangle} > r$.

As for the other exact algorithms, in table 7 we provide a selection of critical values, and the related code is in appendix A.

# 7. Approximations versus exact results

One may want to compare the exact probabilities computed with the new algorithms with the approximations seen in §2. More generally, the operating characteristics of the exact versus the

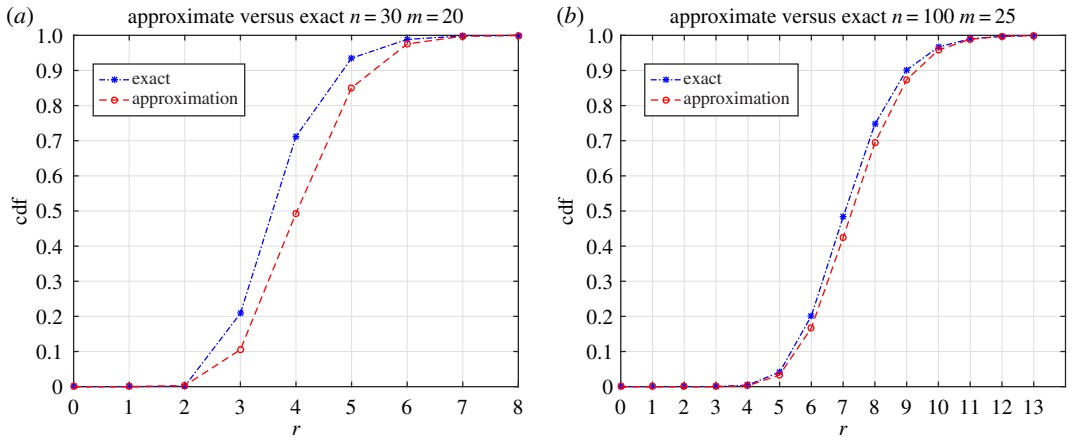

**Figure 2.** Comparison (cdf: cumulative distribution function) between the exact probabilities for the range and the approximation by Young [7] for $n = 30$ and $m = 20$ (*a*), and for $n = 100$ and $m = 25$ (*b*).

approximate tests can be explored. Here, we mainly focus our attention on the multinomial range, given that its use has been proposed in the literature for several statistical tests [2].

In goodness-of-fit tests, under the name 'urn tests' [3], identically independently distributed (i.i.d.) observations generated from a given statistical distribution and properly binned can be described as the result of a multinomial experiment.

Using the approximations given in §2, Young showed that a goodness-of-fit test based on the multinomial range has power advantages with respect to classical alternatives such the $\chi^2$ [7]. For a more general comparison between tests based on the multinomial order statistics and the classical $\chi^2$ goodness-of-fit test statistic, we refer to [14] and [9], respectively.

Consider a sample of i.i.d. observations $X_1, \ldots, X_n$ from a distribution $F$ on the real line, with interest in testing some null hypothesis $\tilde{H}_0 : F = F_0$.

The support of the distribution hypothetical $F_0$ can be partitioned into *m equiprobable* non-overlapping sub-intervals $B_1, \ldots, B_m$ (for the optimal choice of the number of intervals, we refer to [7,15]). We then define the variables $N_1, \ldots, N_m$ as the absolute frequencies of the actual observations in the sample that fall in the intervals $B_1, \ldots, B_m$, that is $N_i = \sum_{j=1}^{n} I(X_j \in B_i)$ for $1 \leq i \leq m$.

By construction of the bins, under $\tilde{H}_0$, the random vector of counts $(N_1, \ldots, N_m)^{\mathrm{T}}$ follows the Mult($n$, **p**) distribution with $\boldsymbol{p} = \boldsymbol{p_0} = (1/m, \ldots, 1/m)^{\mathrm{T}}$. Note that $\tilde{H}_0$ is then transformed into the multinomial null hypothesis $H_0$. Hence we can assess the goodness of fit to $F_0$ by considering the transformed hypothesis testing problem: $H_0 : \boldsymbol{p} = \boldsymbol{p_0}$ versus $H_1 : \boldsymbol{p} \neq \boldsymbol{p_0}$.

To test the multinomial equiprobability hypothesis, we can use the multinomial range as test statistic, as suggested in [7]. In what follows we compare the performances of our exact results, with those of the approximation discussed in §2.

Figure 2 shows two comparisons between the exact and approximate cumulative distribution functions of the multinomial range, in the multinomial experiment, for $n = 30$ and $m = 20$, and for $n = 100$ and $m = 25$. As expected, when the number of balls is small with respect to the number of urns, the approximation is rather poor. Good results are only obtainable for $n \geq 5\,m$. This is in line with the findings in [7].

In our experiments, given the discrete nature of the test statistic, we constructed randomized testing procedures to ensure that the desired level of significance $\alpha$ (0.05) could indeed be achieved exactly in all cases [16].

As the null distribution $F_0$, we used a normal distribution $N(\mu, \sigma^2)$ with $\mu = 1.3$ and $\sigma = 0.25$, while as the alternative distribution $F_1$ we considered a lognormal distribution LN($\mu_{\mathrm{LN}}, \sigma_{\mathrm{LN}}$) with $\mu_{\mathrm{LN}} = 0$ and $\sigma_{\mathrm{LN}} = 0.25$. The two densities are shown in figure 3.

The power of the exact and of the approximate test procedures were computed via Monte Carlo estimation: for each combination of number of bins ($m$), and number of observations ($n$), we generated 3000 i.i.d. samples.

Figure 4 shows the power advantage of the test based on the exact distribution, for all test sample sizes, and for the alternative hypothesis we have considered here. The figure tells that, when using $m = 5$ or 10 urns, the exact test outperforms the approximate one for sample sizes smaller than 25, and it is at least as powerful as the approximate test in the other cases.

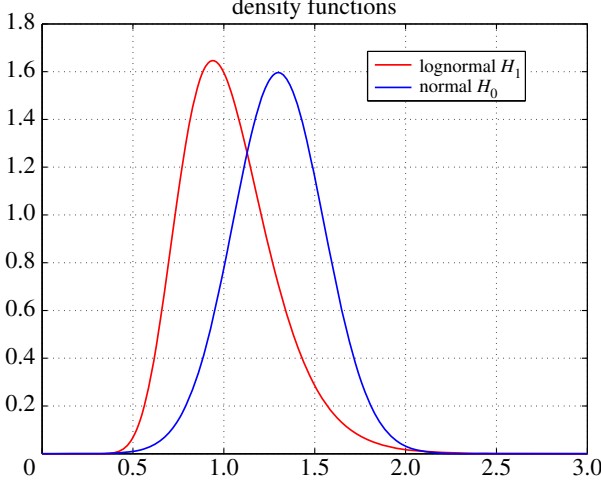

**Figure 3.** Densities of a normal (1.3, 0.25) and of a lognormal (0, 0.25).

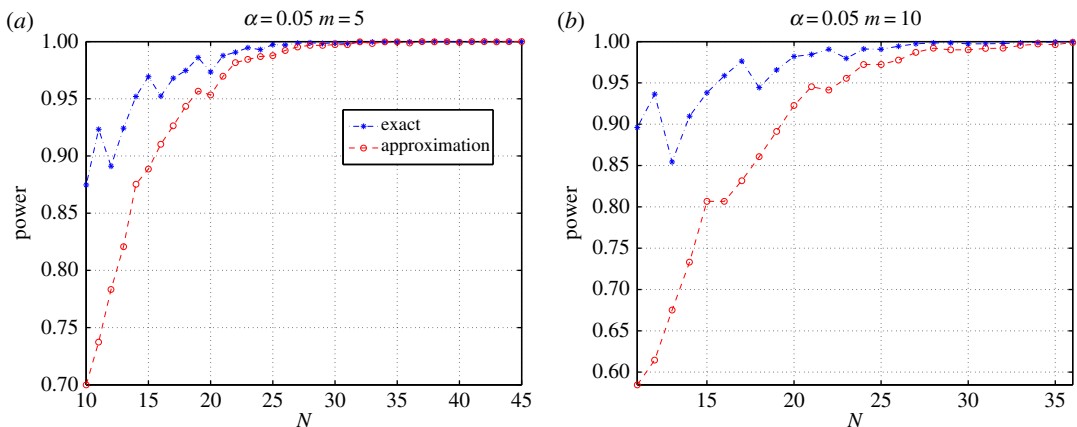

**Figure 4.** Power comparison of the goodness-of-fit tests when using 5 equiprobable bins (*a*) or 10 equiprobable bins (*b*).

Figure 5 compares the powers of the tests for $m = 30$ and $m = 50$, respectively. The figure shows that the exact-distribution-based test is more powerful than the approximate one, for all values of $n$ up until about 50, when all powers approach one.

For both figures 4 and 5, standard errors can be computed using the simple formula $\sqrt{p(1-p)/3000}$, where $p$ is the estimated power.

While this small simulation exercise clearly has no ambition of being exhaustive, on the basis of the findings above, we can say that—as one would expect—the exact test appears to perform better than the approximate one, over a wide range of sample sizes, and for different numbers of urns. Moreover, the improvement in power seems to increase with the number of urns, and it can be very large for the smaller sample sizes. However, one should keep in mind that the observed differences in the performances of the two procedures (exact versus approximation) may also depend upon the accuracy of the calculations of the tail probabilities under the approximation formulae, which may produce type I error probabilities different from the desired ones.

# 8. Uniformly most powerful tests and sums of multinomial counts

We now discuss two constructions of testing procedures for the multinomial probabilities that motivate the use of the largest and of the sum of the $J$ largest multinomial counts.

We consider *one* sample $\mathbf{N} = (N_1, \ldots, N_m)^{\mathrm{T}} = (n_1, \ldots, n_m)^{\mathrm{T}} = \mathbf{n}$, where $\sum_{j=1}^{m} n_j = n$, from the Multinomial($n$, $\mathbf{p}$) distribution with $\mathbf{p} = (p_1, \ldots, p_m)^{\mathrm{T}}$. We focus on the null hypothesis $H_0: p_1 = p_2 = \cdots = p_m = 1/m$, or $H_0: \mathbf{p} = \mathbf{p_0}$ in the notation of §1.

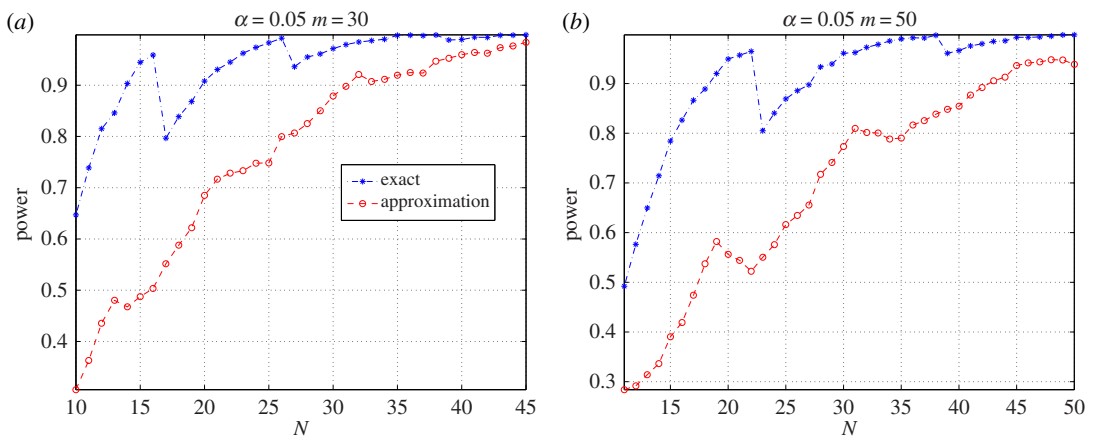

**Figure 5.** Power comparison of the goodness-of-fit tests when using 30 equiprobable bins (*a*) or 50 equiprobable bins (*b*).

In this section, we use the notation $L(\mathbf{p}; \mathbf{n}, m, n)$ for the likelihood function corresponding to a sample of size one from the multinomial distribution (i.e. its pmf).

Similar arguments to those that follow, with the appropriate changes, can be exploited to motivate the use of the minimum (or the sum of the $J$ smallest) counts.

## 8.1. Uniformly most powerful test for an increase in one probability

First, note that if we restrict all the $p_j$ for $j \neq i$ to be equal (and still $\sum_{j=1}^{m} p_j = 1$), then for a given value of $p_i$ one necessarily has $p_j = (1 - p_i)/(m - 1)$, for all $j \neq i$. Now, consider testing the null hypothesis $H_0$ of equiprobabilty against the alternative hypothesis that corresponds to an increased probability of attraction for the $i$th urn, with all other probabilities being equal:

$$H_{1i}: p_i = p_+ > \frac{1}{m}, \quad p_j = \frac{1 - p_+}{m - 1} \quad \forall j \neq i,$$

where $p_+ \in (1/m, 1)$. The most powerful (MP) level $\alpha$ test ($\alpha \in (0, 1)$) for the problem $H_0$ versus $H_{1i}$ can be easily obtained by direct application of the Neyman–Pearson's lemma (e.g. [17]). The rejection region of the MP test is defined by

$$\frac{L(\mathbf{p} = (\frac{1}{m}, \ldots, \frac{1}{m}); \mathbf{n}, m, n)}{L(p_+; \mathbf{n}, m, n)} = \frac{\dfrac{n!}{n_1! \cdots n_m!} \left(\dfrac{1}{m}\right)^n}{\dfrac{n!}{n_1! \cdots n_m!} p_+^{n_i} \left(\dfrac{1 - p_+}{m - 1}\right)^{n - n_i}}$$

$$= \left[\frac{m - 1}{m(1 - p_+)}\right]^n \left[\frac{1 - p_+}{p_+(m - 1)}\right]^{n_i}$$

being less than or equal to some constant such that the probability of rejection under the null be equal to $\alpha$ (note the slightly inconsistent—but convenient—notation for $L(p_+; \mathbf{n}, m, n)$ versus $L(\mathbf{p}; \mathbf{n}, m, n)$). Since $p_+ > 1/m$, it follows immediately that $(1 - p_+) < (m - 1)p_+$, so that the rejection region can be written equivalently as $N_i \geq k_\alpha$. Given the discrete nature of the multinomial random vector, the rejection region must be augmented by construction of a randomization procedure to allow the test to have size $\alpha$.

Since this rejection region does not depend on the choice of $p_+$, the MP test is actually uniformly most powerful (UMP) for the wider testing problem

$$H_0: p_1 = \cdots = p_m = \frac{1}{m} \quad \text{versus} \quad H_{1i}: p_i > \frac{1}{m}, \quad p_j = \frac{1 - p_i}{m - 1} \quad \forall j \neq i.$$

Notice that this same rejection region emerges as the UMP test if one considers just the $i$th marginal count $N_i$, which is distributed as a Binomial($n$, $p_i$) random variable, and the alternative testing problem $H_{0i}^*: p_i = 1/m$ versus $H_{1i}^*: p_i > 1/m$.

Let us now combine the collection of such tests obtained by letting $i = 1, \ldots, m$. Within the parameter space $\Theta = \{p_i \in (0, 1), i = 1, \ldots, m: \sum_{i=1}^{m} p_i = 1\}$, the null hypothesis $H_0$ can be written equivalently as $H_0 : \mathbf{p} \in \Theta_0 = \{1/m, \ldots, 1/m\}$. Trivially, $\Theta_0 = \bigcap_{i=1}^{m} \Theta_0$, so we may construct a global union-intersection

test that rejects $H_0$ whenever at least one of the $m$ tests rejects (e.g. [18]). Easily, such test would then have a rejection region of the form

$$\bigcup_{i=1}^{m}\{N_i \geq k^*\} = \{\max(N_1, \ldots, N_m) \geq k^*\}, \tag{8.1}$$

where $k^*$ is common to the $m$ tests due to the symmetric nature of the individual tests over $i = 1, \ldots, m$ under the null. To control the overall type I error probability $\alpha$, the test should therefore reject if and only if $\max(N_1, \ldots, N_m) \geq k^*$, where the constant $k^*$ should satisfy the size requirement that $P(\max(N_1, \ldots, N_m) \geq k^*; H_0) = \alpha$, and it should therefore be obtained from the distribution of the test statistic $N_{\langle 1 \rangle} = \max(N_1, \ldots, N_m)$ under $H_0$. Observe that, in practice, the $m$ individual tests (and the global test) are randomized, so that (8.1) holds only approximately. Nevertheless, the construction provides quite a strong motivation for the use of the test based on the largest observed count.

## 8.2. Uniformly most powerful test for an equal increase in two probabilities

Let us now consider the case of an equal increase of *two* of the $m \geq 3$ attraction probabilities, corresponding to the two urns $i$ and $j$, $i \neq j$. The level $\alpha$ UMP test for the problem

$$H_0 \text{ versus } H_{1ij} : p_i = p_j = p_+ > \frac{1}{m}, \quad p_h = \frac{1 - 2p_+}{m - 2} \quad \forall h \neq i, j$$

can also be obtained from the Neyman–Pearson's lemma. Note that $p_+ < 0.5$ must hold.

The likelihood function[2] corresponding to the two probabilities $p_i$ and $p_j$ being the same, with all the others being equal, is

$$L(p_+; \mathbf{n}, m, n) = \frac{n!}{n_1! \cdots n_m!} p_+^{n_i} p_+^{n_j} \left( \frac{1 - 2p_+}{m - 2} \right)^{n - (n_i + n_j)}.$$

And the MP test rejects if and only if the likelihood ratio

$$\frac{L(\frac{1}{m}; \mathbf{n}, m, n)}{L(p_+; \mathbf{n}, m, n)} = \frac{\dfrac{n!}{n_1! \cdots n_m!} \left( \dfrac{1}{m} \right)^n}{\dfrac{n!}{n_1! \cdots n_m!} p_+^{n_i + n_j} \left( \dfrac{1 - 2p_+}{m - 2} \right)^{n - (n_i + n_j)}}$$

is less than or equal to some constant. It is easy to see that since $p_+ > (1 - 2p_+)/(m - 2)$, the rejection region can be written equivalently as $N_i + N_j \geq k$ (again with randomization adjustment). Since the test does not depend on the value of $p_+$ as long as it is greater than $1/m$ (and smaller than 0.5), the same test is UMP level $\alpha$ for the testing problem $H_0$ versus $H_{ij} : p_i = p_j > 1/m$, $p_h = (1 - 2p_i)/(m - 2) \, \forall h \neq i, j$.

For this case, one should note that the test that we have derived would *not* be UMP if we allowed $p_i$ and $p_j$ to take different values (both greater than $1/m$, with sum less than one) under the alternative hypothesis $\tilde{H}_{ij}$. Indeed, simple calculations show that, in such a case, the rejection region of the MP test for $H_0$ versus $\tilde{H}_{ij} : p_i = \tilde{p}_i, p_j = \tilde{p}_j, \; p_h = (1 - \tilde{p}_i - \tilde{p}_j)/(m - 2) \, \forall h \neq i, j$ would look like $\gamma_i^{N_i} \gamma_j^{N_j} > k$, where $\gamma_i = \tilde{p}_i(m - 2)/(1 - \tilde{p}_i - \tilde{p}_j) > 1$, and similarly for $\gamma_j$. The (randomized) MP level $\alpha$ test would therefore require the rejection threshold $k$ to be obtained from the distribution, under $H_0$, of the test statistic $\gamma_i^{N_i} + \gamma_j^{N_j}$, or equivalently of the test statistic $N_i \log(\gamma_i) + N_j \log(\gamma_j)$. As a consequence, the rejection threshold would depend on the specific values $\tilde{p}_i$ and $\tilde{p}_j$, and the resulting test could therefore *not* be UMP for the wider testing problem $H_0$ versus $H_{ij}^* : p_i > 1/m, p_j > 1/m, \; p_h = (1 - p_i - p_j)/(m - 2) \, \forall h \neq i, j$.

On the other hand, if we restrict ourselves to the case $p_i = p_j$, then we can again generalize this UMP level $\alpha$ test to reject against the alternative hypothesis that, for *some* pair $(i, j)$ of probabilities, there has been an equal increase from $1/m$. Here, too, the $\binom{m}{2}$ UMP level $\alpha$ tests can be combined—up to the mentioned approximation due to the randomized nature of the tests—by rejecting $H_0$ whenever at least one among them does. The resulting rejection region is then

$$\bigcup_{i \neq j}\{N_i + N_j \geq k\} = \{N_{\langle 1 \rangle} + N_{\langle 2 \rangle} \geq k\},$$

where the equality of the two rejection regions can be easily verified. Again, the constant $k$ that would ensure the overall type I error probability $\alpha$ should be obtained from the distribution of the sum of the two largest observed multinomial counts $N_{\langle 1 \rangle} + N_{\langle 2 \rangle}$.

The discussion in this subsection can be easily extended to the case of an equal increase in more than two of the $m$ probabilities.

---

[2]For ease of notation, we still call it $L(p_+; \mathbf{n}, m, n)$.

**Table 1.** Power of the exact and approximate test in the case of Poisson processes with intensity function $\lambda(t)$.

| intensity function | parameters | exact test | approximate test |
|---|---|---|---|
| $\lambda(t) = 2 + 0.01 * t$ | $m = 30$ and $T = 20$ | 0.184 | 0.037 |
| $\lambda(t) = 0.3 * t$ | $m = 20$ and $T = 200$ | 0.427 | 0.371 |
| $\lambda(t) = 0.05 * t$ | $m = 15$ and $T = 20$ | 0.133 | 0.059 |
| $\lambda(t) = 2 + \sin(2\pi t)$ | $m = 30$ and $T = 200$ | 0.213 | 0.084 |

# 9. Two illustrations

## 9.1. Test for the homogeneity of a Poisson process

Suppose $M(t)$ is a homogeneous Poisson process on the line. If we partition its time domain into $m$ non-overlapping equal-length sub-intervals $B_1, \ldots, B_m$, then, conditionally on the total number of events observed $n$, the numbers of events $N_i$ in the intervals $B_i$, $i = 1, \ldots, m$, follow an equiprobable multinomial distribution [2]. Thus, a test for the homogeneity of the Poisson process is readily constructed from the absolute frequencies.

We simulate a non-homogeneous Poisson process (NHPP) and check whether the multinomial range test is able to identify the non-homogeneity. We use the time-scale transformation of a homogeneous Poisson process (HPP) with (constant) rate equal to one to generate the desired NHPP.

The inter-arrival times $T$ of an HPP with rate one are known to be exponentially distributed with intensity one, i.e. $P(T \geq t) = \exp(-t)$. The inter-arrival times $T'$ for an NHPP are such that $P(T' \geq t) = \exp(-\Lambda(t))$, where $\Lambda(x)$ is the integrated rate function of the process, i.e. the expected number of points in the interval $(0, x]$, with $\Lambda(0) = 0$ (this is also called the cumulative hazard function). It is immediate to see that if $T_1, \ldots, T_n$ are a sample of inter-arrival times generated from the HPP with rate one, then the transformed times $T'_i = \Lambda^{-1}(T_i)$, $i = 1, \ldots, n$, are a sample from an NHPP with integrated rate function $\Lambda(t)$. Hence we can simulate an NHPP by simply sampling exponential variables with parameter equal to one, and by taking the inverse $\Lambda^{-1}$ of the generated inter-arrival times.

Four examples of power comparison (based on 1000 simulated samples each) between the exact and the asymptotic test are shown in table 1, where three linear assumptions and a sine-shaped one for the NHPP intensity function are considered, for different numbers of bins ($m$) and maximum overall time length ($T$). Type I error probability was set equal to 0.05. One can easily appreciate the evident advantage of the exact test over the asymptotic one.

Let us focus a bit more on the effect of choosing different numbers of disjoint intervals, when testing for the homogeneity of a Poisson process. Assume that the true intensity function is described by the harmonic function $\lambda(t) = 2 + \sin(5\pi t)$. Figure 6 shows the power of the two tests when the time domain ($T = 200$) is split into 10 or 20 disjoint intervals. The blue and the red lines represent the case with 10 intervals, whereas the black and the green the one with 20. We can observe once again the better power performances of the exact test. The huge power gain between the two cases with 10 and 20 bins can be explained by the following fact: if we split the time domain into 20 non-overlapping, equal-length intervals, each of them covers the period of the intensity function, averaging out the effect of the harmonic function.

Therefore, the exact range-based test shows better performances with respect to the approximate one, when dealing with Poisson processes, but the specific performance may be quite dependent on the width (i.e. the number) of the bins.

## 9.2. An application to disease clustering

We now discuss a simple application of the multinomial range test to the problem of disease clusters' detection, something frequently of interest to epidemiologists and biostatisticians [19,20].

Often, disease clustering is initially approached as a hypothesis testing problem. The main goal is to test a null hypothesis of no clustering, i.e. a common rate of disease across the study region, against an alternative hypothesis of presence of clusters, or more generally of deviations from the null spatial distribution.

Numerous ways to construct such tests were proposed over the years, and one of the possible approaches is based on the multinomial distribution. In this case, the study region is divided into (roughly) equal population subregions. Then, under the null hypothesis of a common rate of disease across the wider

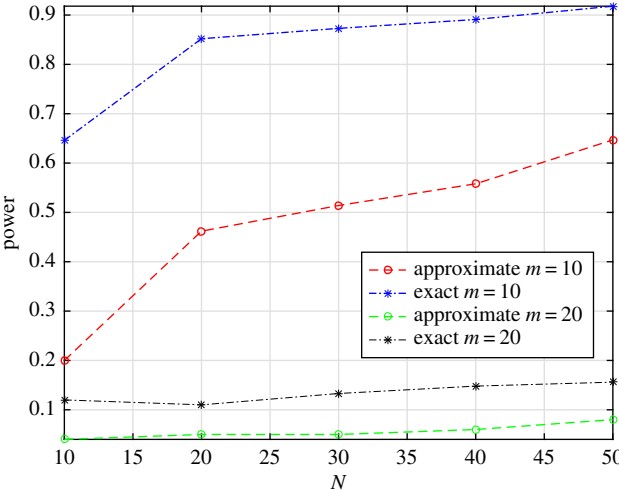

**Figure 6.** Power of the tests in the case of harmonic intensity function $\lambda(t) = 2 + \sin(5\pi t)$ for the NHPP.

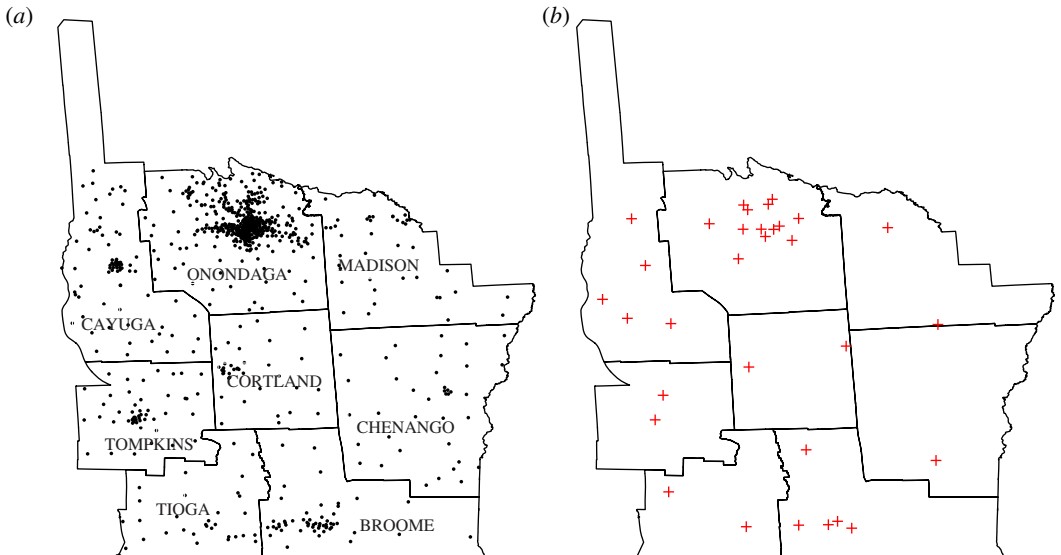

**Figure 7.** Distribution of leukaemia cases over the eight counties of the state of New York (*a*), and positions of the centroids of the 32 new subregions (*b*).

region, and conditionally on the total number of cases observed in the region, the number of events (cases) observed in the different subregions follows an equiprobable multinomial distribution.

We briefly illustrate this approach by using a well-known epidemiological dataset of diagnosed leukaemia cases over eight counties in upstate New York [20]. These data originated from the New York State Cancer Registry, and were gathered during the 5-year period 1978–1982, with a total of 584 individuals diagnosed with leukaemia over a population of approximately 1 million people. The original data contain spatial information about registered events split into 790 census tracts, which, however, have different population sizes. Their spatial distribution is shown in figure 7*a*.

In order to perform the multinomial tests based on the range, we have to group the data points into subregions of approximately equal population. Ideally, the population of these subregions should be exactly the same, but that is impossible due to the original grouping in census tracts. To the best of our knowledge, there is no existing unique algorithm to create spatial partitions such that the elements of the partition have equal population. We propose the following (admittedly ad hoc) procedure:

(i) Define the number of the new subregions to be constructed.
(ii) Use that number to compute a desired value for the population of each subregion.

**Table 2.** Number of subjects (Nr) diagnosed with leukaemia in each of 32 subregions (Subr).

| Subr | Nr | Subr | Nr | Subr | Nr | Subr | Nr |
|------|----|------|----|------|----|------|----|
| 1 | 34 | 9 | 18 | 17 | 25 | 25 | 14 |
| 2 | 28 | 10 | 14 | 18 | 20 | 26 | 12 |
| 3 | 13 | 11 | 17 | 19 | 9 | 27 | 24 |
| 4 | 23 | 12 | 39 | 20 | 21 | 28 | 17 |
| 5 | 23 | 13 | 20 | 21 | 24 | 29 | 5 |
| 6 | 20 | 14 | 17 | 22 | 12 | 30 | 4 |
| 7 | 18 | 15 | 14 | 23 | 31 | 31 | 3 |
| 8 | 27 | 16 | 13 | 24 | 24 | 32 | 3 |

(iii) Create subregions, roughly satisfying the desired population size, around locations with high initial population. Specifically, use the k-means algorithm to create subregions for the rest of the observations, initializing the subregions with locations that have the highest populations.

(iv) Trade observations between subregions based on the population and the distance, until the population constraints are satisfied.

Using this approach, we were able to create new subregions with approximately equal population (some of the points were reassigned to a different subregion in the post-processing stage). In figure 7*b*, 32 subregion centroids are shown. For each subregion, the population varies from 36 036 to 39 528 (roughly 3.48% of the whole population). Under the null hypothesis, these new subregions generate cases with (roughly) the same intensity.

In this grouping, the maximum number of individuals diagnosed with leukaemia within one subregion is 39 versus a minimum of three cases, so that the multinomial range statistic is equal to 33. The total number of cases registered in each subregion is presented in table 2.

Using the exact test based on the range of the sample, the null hypothesis of equiprobable multinomial distribution is rejected at the 0.01 level, since the 99th percentile of the range's exact distribution in the case of 584 events and 32 bins is 25. The null hypothesis is also rejected when one uses 25 subregions (results not shown), which supports previous research about leukaemia cases for this dataset [20].

Importantly, in this illustration there is no difference in our conclusions between using the test based on the exact or the approximate distribution of the range, since for $n = 584$ and $m = 32$ the approximate distribution is close to the exact one. For smaller datasets, the use of the exact test would, however, be quite preferable.

## 10. Discussion

The use of the order statistics computed from the (clearly dependent) multinomial counts opens the possibility of revising several approaches, which have been followed for many decades to assess deviations from the equal probability null hypothesis. One should expect some of the new test statistics to perform particularly well against some alternatives, for example, when one or a few cells have very high (or very low) probabilities associated with them.

From our small simulation exercise, these test statistics appear to benefit greatly from the use of their exact distributions, avoiding the problems due to the use of approximations. Note that more extensive explorations can be pursued using the code that we are providing in this article.

The exact distributions could also be used to develop additional test statistics, best suited for specific problems. For example, in the disease clustering illustration one could recognize that disease cluster studies often originate from a perceived high risk of disease in some subregion(s). Such investigations then run the risk of falling into what is known as the Texas sharpshooter fallacy, i.e. finding an apparently statistically significant difference in risk, when comparing the disease rate in that subregion to the rest of the region. The fact that one is looking at that comparison conditionally on having observed a high rate, if not taken into account properly, can easily produce false positive errors. Note that in such setting, one is actually comparing exactly the largest multinomial order statistic (or one of the largest ones, or their sum) to the counts observed in the other subregions. Given the conditioning on the total number of cases observed, this is the same as constructing the test on just the largest (or the sum of the few largest) order

statistics from the multinomial vector. Such statistics could then naturally be used to test the null hypothesis of homogeneous distribution of the disease risk. Our algorithms therefore allow one to take into account the selection process of the subregion under investigation, within the appropriate statistical testing framework.

Other interesting applications of the exact tests are in the field of risk management. In particular, in credit risk modelling of the number of defaults of counterparties that share the same creditworthiness in terms of rating, but belong to different industrial sectors [21]. In these settings, given the small number of default events and the large number of sectors for the exposures in a granular portfolio, exact multinomial tests may represent an advantage with respect to the more commonly used $\chi^2$ approximation. A similar reasoning is even more relevant if one restricts her attention on low default portfolios (LPDs) [22].

Data accessibility. The data used for the epidemiological application are available as additional resources to the book *Case Studies in Biometry* [20], and downloadable from https://www.stats.ox.ac.uk/pub/datasets/csb/ (last checked on 4 July 2019). All the other data are based on simulations, for which all the necessary information for reproduction is given.

Authors' contributions. M.B. found the original work by Rappeport [9], proposed the idea of the paper, developed the part about UMP tests and helped in drafting and correcting the manuscript and in testing the codes. P.C. performed the literature review, co-developed the algorithms, worked on the graphical part and drafted the manuscript. A.O. co-developed the algorithms, worked on the two illustrations and wrote the Matlab codes. All authors gave final approval for publication.

Competing interests. We declare we have no competing interests.

Funding. P.C. acknowledges the support of the Marie Curie CIG Multivariate Shocks (PCIG13-GA-2013-618794) from the European Union.

Acknowledgements. The authors thank two anonymous referees for their help in improving the paper.

# Appendix A. Codes and tables

In this appendix, we collect the Matlab code that implement all the exact algorithms discussed in this article, as well as some tables of critical values.

## Algorithm for the Maximum

```
1  function [P] = max_order_statistic(t,n,m)
2  %% Script calculates probability of highest order statistic being <=t
3  % Under equiprobable multinomial
4  % Input :
5  % n — number of balls
6  % m — number of urns(cells)
7  % t — argument of cdf
8  P = 0;
9  if t == 0 && n~=0
10      P = 0;
11      return
12  end
13  if t==0 && n==0
14      P=1;
15      return
16  end
17  if t>=n
18      P = 1;
19      return
20  end
21  if n==0 && m~=0
22      P=1;
23      return
24  end
```

```matlab
25   if n==0 && m==0
26       P=1;
27       return
28   end
29   common_term = gammaln(n+1)+gammaln(m+1)−n*log(m);
30   switch t
31       case 1
32           if m>=n
33               P = exp(gammaln(m+1)−gammaln(m−n+1)−n*log(m)); % explicit
                     calculation of P(n<1><=1)
34               calc = [calc;t,n,m,P];
35           else
36               P = 0;
37               calc = [calc;t,n,m,P];
38           end
39       otherwise
40           % range of summation for q
41           LowSum = max(0,n−t*m+m);
42           UpSum = floor(n/t);
43           for q = LowSum:UpSum
44               summ_term = (−q*gammaln(t+1)−gammaln(q+1)−gammaln(m−q+1)−gammaln(n−t
                     *q+1));
45               if m==q
46                   summ_term_nominator = 0;
47               else
48                   summ_term_nominator = (n−t*q)*log(m−q);
49               end
50               coef = exp(common_term+summ_term+summ_term_nominator);
51                   [temp]  = max_order_statistic(t−1,n−t*q,m−q);
52
53               P = P + coef*temp;
54
55           end
56   end
57   end
```

## Sum of the J Largest Order Statistics

```matlab
1    function [ P ] = highest_order_statistics( t,n,m,J )
2    %% Probability of the sum of the first J largest order statistics being smaller
         than t
3    %   P(sum(1:J)n<i> <= t : n,m) under H1 − equiprobable multinomial
4    % Input:
5    % n− number of trials
6    % m − number of cells(urns)
7    % t − argument of cdf
8    % J − number of largest order statistics
9    if J>m
10       error('J should be smaller or equal than m')
11   end
12   if J==m && (t<n )
13       error(' Total sum is every time equal to n')
14   end
```

```matlab
15
16   if t == 0 && n~=0
17       P = 0;
18       return
19   end
20   if t==0 && n==0
21       P=1;
22       return
23   end
24   if t>=n
25       P = 1;
26       return
27   end
28
29   %% first term if n<1> <= t/J
30   P = max_order_statistic(floor(t/J),n,m);
31   %% recursive summation over all possible options
32   for sum_depth = 1 : J—1
33       rangeArg = [];
34       cur_depth = 1;
35       P = P + recursive_sum(t,n,m,J,sum_depth,cur_depth,rangeArg);
36   end
37   end
38   function S = recursive_sum(t,n,m,J,sum_depth,cur_depth,rangeArg)
39   %% auxiliary function for calculating sum of nested loops
40   S = 0;
41   if cur_depth <= sum_depth % either increment summation depth or calculate the
          term
42       if cur_depth == 1
43           cur_range(1) = floor(t/J+1);
44           cur_range(2) = t—sum_depth+1;
45       else
46           cur_range(1) = floor((t — sum(rangeArg(1:cur_depth—1)))/(J— cur_depth+1)
                  +1);
47           cur_range(2) = min( rangeArg(cur_depth—1), t — sum(rangeArg(1:cur_depth
                  —1)));
48       end
49       for r=cur_range(1):cur_range(2)
50          rangeArg(cur_depth) = r;
51           S = S + recursive_sum(t,n,m,J,sum_depth,cur_depth+1,rangeArg);
52       end
53   else
54       prob_arg = floor( (t—sum(rangeArg))/(J—sum_depth));
55       temp_p = max_order_statistic(prob_arg, n — sum(rangeArg),m—sum_depth);
56       common_term = gammaln(n+1) + gammaln(m+1) — n*log(m);
57       coef = (n—sum(rangeArg))*log(m—sum_depth) — gammaln(m—sum_depth+1) — gammaln
              ( n — sum(rangeArg)+1);
58       for k=1:numel(rangeArg)
59           coef = coef — gammaln(rangeArg(k)+1);
60       end
61       equal_statistics = unique(rangeArg);
62       for k=1:numel(equal_statistics)
```

```
63            temp = numel(find(rangeArg == equal_statistics(k)));
64            coef = coef   - gammaln(temp+1);
65        end
66        S  = temp_p*exp(common_term+coef);
67
68  end
69  end
```

## Algorithm for the Minimum

```
1   function P = smallest_order_value( t,n,m )
2   %% Function to calculate the probability of smallest order statistic to be >=
         than t for equiprobable multinomial
3   % Input:
4   % t  - argument of "survival" function
5   % n - number of balls
6   % m  - number of cells
7
8   P = 0;
9   % add for exceptions for "naive" input
10  if t>floor(n/m)
11      P=0;
12      return
13  end
14  if t==0
15      P=1;
16      return
17  end
18      aux = max_for_min(n,n,m,calc,t);
19      P = P + aux;
20  end
21
22  function [aux,calc] = max_for_min(t_max,n,m,calc,t)
23  aux = 0;
24  if t_max<t
25      if n==0 && m==0
26          aux=1;
27          return
28      else
29          aux = 0;
30          return
31      end
32  else
33      if n==0 && m == 0
34          aux=1;
35          return
36      end
37      if t_max==1
38          if m==n
39              aux = exp(gammaln(m+1)-gammaln(m-n+1)-n*log(m)); % explicit
                     calculation of P(n<1><=1)
40              return
41          else
```

```
42          aux= 0;
43              return
44          end
45       end
46      if n==0 && m~=0
47          aux=0;
48          return
49      end
50      common_term = gammaln(n+1)+gammaln(m+1)−n∗log(m);
51      LowSum = max(0,n−t_max∗m+m);
52      UpSum = floor(n/t_max);
53      for q = LowSum:UpSum
54          summ_term = (−q∗gammaln(t_max+1)−gammaln(q+1)−gammaln(m−q+1)−gammaln(n−
                t_max∗q+1));
55          if m==q
56              summ_term_nominator = 0;
57          else
58              summ_term_nominator = (n−t_max∗q)∗log(m−q);
59          end
60          coef = exp(common_term+summ_term+summ_term_nominator);
61          [temp]  = max_for_min(t_max−1,n−t_max∗q,m−q,t);
62          aux = aux + coef∗temp;
63      end
64  end
65  end
```

## Algorithm for the Range

```
1   function [P] = range_probability( t,n,m )
2   %% Function to calculate the probability of the range to be < =than t
3   %for equiprobable multinomial
4   % Input:
5   % r  − argument of "cdf" function
6   % n − number of balls
7   % m  − number of cells
8   P = 0;
9   t= floor(t);
10  if t>n
11      P=1;
12      return
13  end
14
15  [P]=max_order_statistic(t,n,m);
16  prev=[t,n,m];
17  for t_max = t+1:n
18      [aux] = max_for_range(t_max,n,m,prev,t);
19      P = P + aux;
20      prev = [t_max,n,m];
21  end
22  end
23
24  function [aux] = max_for_range(t_max,n,m,prev,t)
25
```

```matlab
26   aux=0;
27   if [t_max,n,m]==prev
28       aux = 0;
29       return
30   end
31   if prev(1)+1-t_max>t
32       if n==0 && m==0
33           aux=1;
34           return
35       else
36           aux = 0;
37           return
38       end
39   else
40       if n==0 && m == 0
41           aux=1;
42           return
43       end
44       if t_max==1
45           if m==n
46               aux = exp(gammaln(m+1)-gammaln(m-n+1)-n*log(m)); % explicit
                       calculation of P(n<1><=1)
47               return
48           else
49               aux= 0;
50                           return
51           end
52       end
53       if n==0 && m~=0
54           aux=0;
55                   return
56       end
57
58       common_term = gammaln(n+1)+gammaln(m+1)-n*log(m);
59       LowSum = max(0,n-t_max*m+m);
60       UpSum = floor(n/t_max);
61       for q = LowSum:UpSum
62           summ_term = (-q*gammaln(t_max+1)-gammaln(q+1)-gammaln(m-q+1)-gammaln(n-
                   t_max*q+1));
63           if m==q
64               summ_term_nominator = 0;
65           else
66               summ_term_nominator = (n-t_max*q)*log(m-q);
67           end
68           coef = exp(common_term+summ_term+summ_term_nominator);
69           [temp]  = max_for_range(t_max-1,m-t_max*q,m-q,prev,t);
70           aux = aux + coef*temp;
71       end
72   end
73   end
```

**Table 3.** Distribution of the multinomial maximum. For example, in the case with $n = 10$ and $m = 5$, $P(N_{\langle 1 \rangle} \leq 3; n = 10, m = 5) = 0.433$.

| | | t | | | | | | | | | | | | | | |
|---|---|---|---|---|---|---|---|---|---|---|---|---|---|---|---|---|
| n | m | 2 | 3 | 4 | 5 | 6 | 7 | 8 | 9 | 10 | 11 | 12 | 13 | 14 | 15 | 16 |
| 5 | 5 | 0.710 | 0.966 | 0.998 | 1.000 | 1.000 | 1.000 | 1.000 | 1.000 | 1.000 | 1.000 | 1.000 | 1.000 | 1.000 | 1.000 | 1.000 |
| 10 | 5 | 0.012 | 0.433 | 0.836 | 0.968 | 0.996 | 1.000 | 1.000 | 1.000 | 1.000 | 1.000 | 1.000 | 1.000 | 1.000 | 1.000 | 1.000 |
| 15 | 5 | 0.000 | 0.006 | 0.284 | 0.700 | 0.910 | 0.979 | 0.996 | 0.999 | 1.000 | 1.000 | 1.000 | 1.000 | 1.000 | 1.000 | 1.000 |
| 20 | 5 | 0.000 | 0.000 | 0.003 | 0.200 | 0.584 | 0.840 | 0.950 | 0.987 | 0.997 | 0.999 | 1.000 | 1.000 | 1.000 | 1.000 | 1.000 |
| 25 | 5 | 0.000 | 0.000 | 0.000 | 0.002 | 0.147 | 0.490 | 0.769 | 0.913 | 0.972 | 0.992 | 0.998 | 0.999 | 1.000 | 1.000 | 1.000 |
| 30 | 5 | 0.000 | 0.000 | 0.000 | 0.000 | 0.001 | 0.113 | 0.415 | 0.701 | 0.872 | 0.953 | 0.984 | 0.995 | 0.999 | 1.000 | 1.000 |
| 10 | 10 | 0.396 | 0.873 | 0.984 | 0.999 | 1.000 | 1.000 | 1.000 | 1.000 | 1.000 | 1.000 | 1.000 | 1.000 | 1.000 | 1.000 | 1.000 |
| 20 | 10 | 0.000 | 0.127 | 0.603 | 0.889 | 0.976 | 0.996 | 0.999 | 1.000 | 1.000 | 1.000 | 1.000 | 1.000 | 1.000 | 1.000 | 1.000 |
| 30 | 10 | 0.000 | 0.000 | 0.049 | 0.394 | 0.753 | 0.923 | 0.980 | 0.995 | 0.999 | 1.000 | 1.000 | 1.000 | 1.000 | 1.000 | 1.000 |
| 40 | 10 | 0.000 | 0.000 | 0.000 | 0.022 | 0.259 | 0.617 | 0.848 | 0.950 | 0.985 | 0.996 | 0.999 | 1.000 | 1.000 | 1.000 | 1.000 |
| 50 | 10 | 0.000 | 0.000 | 0.000 | 0.000 | 0.011 | 0.173 | 0.498 | 0.765 | 0.908 | 0.968 | 0.990 | 0.997 | 0.999 | 1.000 | 1.000 |
| 60 | 10 | 0.000 | 0.000 | 0.000 | 0.000 | 0.000 | 0.006 | 0.119 | 0.401 | 0.681 | 0.857 | 0.944 | 0.980 | 0.993 | 0.998 | 0.999 |
| 15 | 15 | 0.219 | 0.785 | 0.966 | 0.996 | 1.000 | 1.000 | 1.000 | 1.000 | 1.000 | 1.000 | 1.000 | 1.000 | 1.000 | 1.000 | 1.000 |
| 30 | 15 | 0.000 | 0.037 | 0.432 | 0.813 | 0.955 | 0.991 | 0.998 | 1.000 | 1.000 | 1.000 | 1.000 | 1.000 | 1.000 | 1.000 | 1.000 |
| 45 | 15 | 0.000 | 0.000 | 0.009 | 0.221 | 0.620 | 0.867 | 0.962 | 0.990 | 0.998 | 1.000 | 1.000 | 1.000 | 1.000 | 1.000 | 1.000 |
| 60 | 15 | 0.000 | 0.000 | 0.000 | 0.002 | 0.114 | 0.451 | 0.755 | 0.911 | 0.972 | 0.992 | 0.998 | 1.000 | 1.000 | 1.000 | 1.000 |
| 75 | 15 | 0.000 | 0.000 | 0.000 | 0.000 | 0.001 | 0.061 | 0.321 | 0.638 | 0.844 | 0.942 | 0.981 | 0.994 | 0.998 | 1.000 | 1.000 |
| 90 | 15 | 0.000 | 0.000 | 0.000 | 0.000 | 0.000 | 0.000 | 0.034 | 0.228 | 0.530 | 0.769 | 0.902 | 0.963 | 0.987 | 0.996 | 0.999 |
| 20 | 20 | 0.121 | 0.706 | 0.949 | 0.993 | 0.999 | 1.000 | 1.000 | 1.000 | 1.000 | 1.000 | 1.000 | 1.000 | 1.000 | 1.000 | 1.000 |
| 40 | 20 | 0.000 | 0.011 | 0.309 | 0.742 | 0.933 | 0.986 | 0.997 | 1.000 | 1.000 | 1.000 | 1.000 | 1.000 | 1.000 | 1.000 | 1.000 |
| 60 | 20 | 0.000 | 0.000 | 0.001 | 0.123 | 0.510 | 0.814 | 0.944 | 0.985 | 0.997 | 0.999 | 1.000 | 1.000 | 1.000 | 1.000 | 1.000 |
| 80 | 20 | 0.000 | 0.000 | 0.000 | 0.000 | 0.050 | 0.329 | 0.671 | 0.874 | 0.958 | 0.988 | 0.997 | 0.999 | 1.000 | 1.000 | 1.000 |

(Continued.)

**Table 3.** (*Continued.*)

| n | m | 2 | 3 | 4 | 5 | 6 | 7 | 8 | 9 | 10 | 11 | 12 | 13 | 14 | 15 | 16 |
|---|---|---|---|---|---|---|---|---|---|----|----|----|----|----|----|----|
| 100 | 20 | 0.000 | 0.000 | 0.000 | 0.000 | 0.000 | 0.021 | 0.207 | 0.532 | 0.785 | 0.916 | 0.971 | 0.991 | 0.997 | 0.999 | 1.000 |
| 120 | 20 | 0.000 | 0.000 | 0.000 | 0.000 | 0.000 | 0.000 | 0.010 | 0.129 | 0.411 | 0.689 | 0.862 | 0.945 | 0.980 | 0.993 | 0.998 |
| 25 | 25 | 0.067 | 0.634 | 0.931 | 0.991 | 0.999 | 1.000 | 1.000 | 1.000 | 1.000 | 1.000 | 1.000 | 1.000 | 1.000 | 1.000 | 1.000 |
| 50 | 25 | 0.000 | 0.003 | 0.222 | 0.678 | 0.912 | 0.981 | 0.996 | 0.999 | 1.000 | 1.000 | 1.000 | 1.000 | 1.000 | 1.000 | 1.000 |
| 75 | 25 | 0.000 | 0.000 | 0.000 | 0.069 | 0.419 | 0.764 | 0.926 | 0.980 | 0.995 | 0.999 | 1.000 | 1.000 | 1.000 | 1.000 | 1.000 |
| 100 | 25 | 0.000 | 0.000 | 0.000 | 0.000 | 0.022 | 0.240 | 0.596 | 0.837 | 0.945 | 0.983 | 0.995 | 0.999 | 1.000 | 1.000 | 1.000 |
| 125 | 25 | 0.000 | 0.000 | 0.000 | 0.000 | 0.000 | 0.007 | 0.133 | 0.443 | 0.730 | 0.891 | 0.961 | 0.987 | 0.996 | 0.999 | 1.000 |
| 150 | 25 | 0.000 | 0.000 | 0.000 | 0.000 | 0.000 | 0.000 | 0.003 | 0.073 | 0.319 | 0.617 | 0.823 | 0.928 | 0.973 | 0.991 | 0.997 |
| 30 | 30 | 0.037 | 0.570 | 0.914 | 0.988 | 0.999 | 1.000 | 1.000 | 1.000 | 1.000 | 1.000 | 1.000 | 1.000 | 1.000 | 1.000 | 1.000 |
| 60 | 30 | 0.000 | 0.001 | 0.159 | 0.619 | 0.891 | 0.975 | 0.995 | 0.999 | 1.000 | 1.000 | 1.000 | 1.000 | 1.000 | 1.000 | 1.000 |
| 90 | 30 | 0.000 | 0.000 | 0.000 | 0.039 | 0.345 | 0.717 | 0.908 | 0.975 | 0.994 | 0.999 | 1.000 | 1.000 | 1.000 | 1.000 | 1.000 |
| 120 | 30 | 0.000 | 0.000 | 0.000 | 0.000 | 0.010 | 0.175 | 0.529 | 0.802 | 0.931 | 0.979 | 0.994 | 0.998 | 1.000 | 1.000 | 1.000 |
| 150 | 30 | 0.000 | 0.000 | 0.000 | 0.000 | 0.000 | 0.003 | 0.086 | 0.369 | 0.678 | 0.866 | 0.952 | 0.984 | 0.995 | 0.999 | 1.000 |
| 180 | 30 | 0.000 | 0.000 | 0.000 | 0.000 | 0.000 | 0.000 | 0.001 | 0.042 | 0.248 | 0.553 | 0.786 | 0.911 | 0.967 | 0.988 | 0.996 |
| 40 | 40 | 0.011 | 0.459 | 0.880 | 0.982 | 0.998 | 1.000 | 1.000 | 1.000 | 1.000 | 1.000 | 1.000 | 1.000 | 1.000 | 1.000 | 1.000 |
| 80 | 40 | 0.000 | 0.000 | 0.081 | 0.516 | 0.850 | 0.965 | 0.993 | 0.999 | 1.000 | 1.000 | 1.000 | 1.000 | 1.000 | 1.000 | 1.000 |
| 120 | 40 | 0.000 | 0.000 | 0.000 | 0.012 | 0.233 | 0.631 | 0.873 | 0.964 | 0.991 | 0.998 | 1.000 | 1.000 | 1.000 | 1.000 | 1.000 |
| 160 | 40 | 0.000 | 0.000 | 0.000 | 0.000 | 0.002 | 0.093 | 0.417 | 0.737 | 0.905 | 0.970 | 0.991 | 0.998 | 0.999 | 1.000 | 1.000 |
| 200 | 40 | 0.000 | 0.000 | 0.000 | 0.000 | 0.000 | 0.000 | 0.035 | 0.256 | 0.585 | 0.819 | 0.932 | 0.977 | 0.993 | 0.998 | 0.999 |
| 50 | 50 | 0.003 | 0.370 | 0.848 | 0.976 | 0.997 | 1.000 | 1.000 | 1.000 | 1.000 | 1.000 | 1.000 | 1.000 | 1.000 | 1.000 | 1.000 |
| 100 | 50 | 0.000 | 0.000 | 0.042 | 0.430 | 0.811 | 0.954 | 0.991 | 0.998 | 1.000 | 1.000 | 1.000 | 1.000 | 1.000 | 1.000 | 1.000 |
| 150 | 50 | 0.000 | 0.000 | 0.000 | 0.004 | 0.157 | 0.556 | 0.840 | 0.953 | 0.988 | 0.997 | 0.999 | 1.000 | 1.000 | 1.000 | 1.000 |
| 200 | 50 | 0.000 | 0.000 | 0.000 | 0.000 | 0.000 | 0.049 | 0.329 | 0.677 | 0.879 | 0.961 | 0.989 | 0.997 | 0.999 | 1.000 | 1.000 |

**Table 4.** Distribution of the sum of the first two order statistics: $N_{\langle 1 \rangle} + N_{\langle 2 \rangle}$. For example, in the case with $n = 10$ and $m = 10$, $P(N_{\langle 1 \rangle} + N_{\langle 2 \rangle} \le 5; n = 10, m = 10) = 0.811$.

| | | t | | | | | | | | | | | | | | | | | |
|---|---|---|---|---|---|---|---|---|---|---|---|---|---|---|---|---|---|---|---|
| n | m | 5 | 6 | 7 | 8 | 9 | 10 | 11 | 12 | 13 | 14 | 15 | 16 | 17 | 18 | 19 | 20 | 21 | 22 |
| 10 | 5 | 0.166 | 0.588 | 0.887 | 0.984 | 0.999 | 1.000 | 1.000 | 1.000 | 1.000 | 1.000 | 1.000 | 1.000 | 1.000 | 1.000 | 1.000 | 1.000 | 1.000 | 1.000 |
| 15 | 5 | 0.000 | 0.006 | 0.088 | 0.392 | 0.722 | 0.913 | 0.981 | 0.997 | 1.000 | 1.000 | 1.000 | 1.000 | 1.000 | 1.000 | 1.000 | 1.000 | 1.000 | 1.000 |
| 20 | 5 | 0.000 | 0.000 | 0.000 | 0.003 | 0.054 | 0.277 | 0.580 | 0.817 | 0.939 | 0.984 | 0.997 | 1.000 | 1.000 | 1.000 | 1.000 | 1.000 | 1.000 | 1.000 |
| 25 | 5 | 0.000 | 0.000 | 0.000 | 0.000 | 0.000 | 0.002 | 0.037 | 0.205 | 0.469 | 0.720 | 0.881 | 0.959 | 0.988 | 0.997 | 0.999 | 1.000 | 1.000 | 1.000 |
| 30 | 5 | 0.000 | 0.000 | 0.000 | 0.000 | 0.000 | 0.000 | 0.000 | 0.001 | 0.027 | 0.157 | 0.385 | 0.631 | 0.816 | 0.923 | 0.972 | 0.992 | 0.998 | 1.000 |
| 10 | 10 | 0.811 | 0.967 | 0.997 | 1.000 | 1.000 | 1.000 | 1.000 | 1.000 | 1.000 | 1.000 | 1.000 | 1.000 | 1.000 | 1.000 | 1.000 | 1.000 | 1.000 | 1.000 |
| 20 | 10 | 0.001 | 0.131 | 0.415 | 0.736 | 0.913 | 0.978 | 0.996 | 0.999 | 1.000 | 1.000 | 1.000 | 1.000 | 1.000 | 1.000 | 1.000 | 1.000 | 1.000 | 1.000 |
| 30 | 10 | 0.000 | 0.000 | 0.050 | 0.050 | 0.206 | 0.494 | 0.745 | 0.898 | 0.966 | 0.991 | 0.998 | 1.000 | 1.000 | 1.000 | 1.000 | 1.000 | 1.000 | 1.000 |
| 40 | 10 | 0.000 | 0.000 | 0.000 | 0.000 | 0.000 | 0.023 | 0.109 | 0.325 | 0.573 | 0.779 | 0.903 | 0.963 | 0.988 | 0.996 | 0.999 | 1.000 | 1.000 | 1.000 |
| 50 | 10 | 0.000 | 0.000 | 0.000 | 0.000 | 0.000 | 0.000 | 0.000 | 0.011 | 0.061 | 0.216 | 0.432 | 0.653 | 0.816 | 0.915 | 0.965 | 0.987 | 0.996 | 0.999 |
| 60 | 10 | 0.000 | 0.000 | 0.000 | 0.000 | 0.000 | 0.000 | 0.000 | 0.000 | 0.000 | 0.006 | 0.036 | 0.148 | 0.325 | 0.538 | 0.721 | 0.852 | 0.929 | 0.969 |
| 15 | 15 | 0.592 | 0.884 | 0.978 | 0.997 | 1.000 | 1.000 | 1.000 | 1.000 | 1.000 | 1.000 | 1.000 | 1.000 | 1.000 | 1.000 | 1.000 | 1.000 | 1.000 | 1.000 |
| 30 | 15 | 0.000 | 0.037 | 0.172 | 0.501 | 0.772 | 0.922 | 0.978 | 0.995 | 0.999 | 1.000 | 1.000 | 1.000 | 1.000 | 1.000 | 1.000 | 1.000 | 1.000 | 1.000 |
| 45 | 15 | 0.000 | 0.000 | 0.000 | 0.009 | 0.052 | 0.251 | 0.511 | 0.749 | 0.893 | 0.962 | 0.988 | 0.997 | 0.999 | 1.000 | 1.000 | 1.000 | 1.000 | 1.000 |
| 60 | 15 | 0.000 | 0.000 | 0.000 | 0.000 | 0.000 | 0.002 | 0.018 | 0.127 | 0.316 | 0.561 | 0.760 | 0.888 | 0.954 | 0.983 | 0.994 | 0.998 | 1.000 | 1.000 |
| 75 | 15 | 0.000 | 0.000 | 0.000 | 0.000 | 0.000 | 0.000 | 0.000 | 0.001 | 0.007 | 0.067 | 0.193 | 0.405 | 0.616 | 0.786 | 0.894 | 0.953 | 0.981 | 0.993 |
| 90 | 15 | 0.000 | 0.000 | 0.000 | 0.000 | 0.000 | 0.000 | 0.000 | 0.000 | 0.000 | 0.000 | 0.003 | 0.037 | 0.119 | 0.287 | 0.485 | 0.674 | 0.815 | 0.905 |
| 20 | 20 | 0.410 | 0.788 | 0.948 | 0.991 | 0.999 | 1.000 | 1.000 | 1.000 | 1.000 | 1.000 | 1.000 | 1.000 | 1.000 | 1.000 | 1.000 | 1.000 | 1.000 | 1.000 |
| 40 | 20 | 0.000 | 0.011 | 0.066 | 0.339 | 0.635 | 0.852 | 0.952 | 0.987 | 0.997 | 0.999 | 1.000 | 1.000 | 1.000 | 1.000 | 1.000 | 1.000 | 1.000 | 1.000 |
| 60 | 20 | 0.000 | 0.000 | 0.000 | 0.001 | 0.012 | 0.131 | 0.340 | 0.610 | 0.808 | 0.922 | 0.972 | 0.991 | 0.998 | 0.999 | 1.000 | 1.000 | 1.000 | 1.000 |
| 80 | 20 | 0.000 | 0.000 | 0.000 | 0.000 | 0.000 | 0.000 | 0.003 | 0.052 | 0.169 | 0.396 | 0.622 | 0.801 | 0.908 | 0.963 | 0.986 | 0.995 | 0.999 | 1.000 |
| 100 | 20 | 0.000 | 0.000 | 0.000 | 0.000 | 0.000 | 0.000 | 0.000 | 0.000 | 0.001 | 0.022 | 0.084 | 0.247 | 0.451 | 0.658 | 0.811 | 0.907 | 0.959 | 0.983 |

(Continued.)

**Table 4.** (*Continued.*)

| t | | | | | | | | | | | | | | | | | | | |
| --- | --- | --- | --- | --- | --- | --- | --- | --- | --- | --- | --- | --- | --- | --- | --- | --- | --- | --- | --- |
| n | m | 5 | 6 | 7 | 8 | 9 | 10 | 11 | 12 | 13 | 14 | 15 | 16 | 17 | 18 | 19 | 20 | 21 | 22 |
| 25 | 25 | 0.273 | 0.695 | 0.912 | 0.982 | 0.997 | 1.000 | 1.000 | 1.000 | 1.000 | 1.000 | 1.000 | 1.000 | 1.000 | 1.000 | 1.000 | 1.000 | 1.000 | 1.000 |
| 50 | 25 | 0.000 | 0.003 | 0.024 | 0.233 | 0.515 | 0.779 | 0.919 | 0.976 | 0.994 | 0.999 | 1.000 | 1.000 | 1.000 | 1.000 | 1.000 | 1.000 | 1.000 | 1.000 |
| 30 | 30 | 0.176 | 0.612 | 0.873 | 0.971 | 0.995 | 0.999 | 1.000 | 1.000 | 1.000 | 1.000 | 1.000 | 1.000 | 1.000 | 1.000 | 1.000 | 1.000 | 1.000 | 1.000 |
| 60 | 30 | 0.000 | 0.001 | 0.008 | 0.163 | 0.412 | 0.708 | 0.884 | 0.963 | 0.990 | 0.998 | 0.999 | 1.000 | 1.000 | 1.000 | 1.000 | 1.000 | 1.000 | 1.000 |
| 40 | 40 | 0.070 | 0.478 | 0.790 | 0.945 | 0.989 | 0.998 | 1.000 | 1.000 | 1.000 | 1.000 | 1.000 | 1.000 | 1.000 | 1.000 | 1.000 | 1.000 | 1.000 | 1.000 |
| 80 | 40 | 0.000 | 0.000 | 0.001 | 0.082 | 0.257 | 0.579 | 0.808 | 0.932 | 0.979 | 0.995 | 0.999 | 1.000 | 1.000 | 1.000 | 1.000 | 1.000 | 1.000 | 1.000 |
| 50 | 50 | 0.026 | 0.378 | 0.708 | 0.915 | 0.981 | 0.997 | 0.999 | 1.000 | 1.000 | 1.000 | 1.000 | 1.000 | 1.000 | 1.000 | 1.000 | 1.000 | 1.000 | 1.000 |
| 100 | 50 | 0.000 | 0.000 | 0.000 | 0.042 | 0.156 | 0.471 | 0.733 | 0.897 | 0.967 | 0.991 | 0.998 | 1.000 | 1.000 | 1.000 | 1.000 | 1.000 | 1.000 | 1.000 |

**Table 5.** Distribution of the sum of the first three largest order statistics: $N_{\langle 1 \rangle} + N_{\langle 2 \rangle} + N_{\langle 3 \rangle}$. For example, in the case with $n = 15$ and $m = 15$, $P\left(\sum_{i=1}^{3} N_{\langle i \rangle} \leq 8; n = 15, m = 15\right) = 0.869$.

| n | m | 8 | 9 | 10 | 11 | 12 | 13 | 14 | 15 | 16 | 18 | 20 | 22 | 24 | 26 | 28 | 30 | 32 | 34 |
|---|---|---|---|---|---|---|---|---|---|---|---|---|---|---|---|---|---|---|---|
| 10 | 10 | 0.987 | 0.999 | 1.000 | 1.000 | 1.000 | 1.000 | 1.000 | 1.000 | 1.000 | 1.000 | 1.000 | 1.000 | 1.000 | 1.000 | 1.000 | 1.000 | 1.000 | 1.000 |
| 20 | 10 | 0.020 | 0.174 | 0.468 | 0.747 | 0.913 | 0.978 | 0.996 | 0.999 | 1.000 | 1.000 | 1.000 | 1.000 | 1.000 | 1.000 | 1.000 | 1.000 | 1.000 | 1.000 |
| 30 | 10 | 0.000 | 0.000 | 0.000 | 0.005 | 0.064 | 0.230 | 0.472 | 0.708 | 0.869 | 0.986 | 0.999 | 1.000 | 1.000 | 1.000 | 1.000 | 1.000 | 1.000 | 1.000 |
| 40 | 10 | 0.000 | 0.000 | 0.000 | 0.000 | 0.000 | 0.000 | 0.002 | 0.028 | 0.120 | 0.509 | 0.856 | 0.977 | 0.998 | 1.000 | 1.000 | 1.000 | 1.000 | 1.000 |
| 50 | 10 | 0.000 | 0.000 | 0.000 | 0.000 | 0.000 | 0.000 | 0.000 | 0.000 | 0.000 | 0.014 | 0.181 | 0.559 | 0.858 | 0.972 | 0.996 | 1.000 | 1.000 | 1.000 |
| 60 | 10 | 0.000 | 0.000 | 0.000 | 0.000 | 0.000 | 0.000 | 0.000 | 0.000 | 0.000 | 0.000 | 0.000 | 0.040 | 0.254 | 0.610 | 0.868 | 0.970 | 0.995 | 0.999 |
| 15 | 15 | 0.869 | 0.972 | 0.996 | 1.000 | 1.000 | 1.000 | 1.000 | 1.000 | 1.000 | 1.000 | 1.000 | 1.000 | 1.000 | 1.000 | 1.000 | 1.000 | 1.000 | 1.000 |
| 30 | 15 | 0.000 | 0.038 | 0.174 | 0.405 | 0.669 | 0.856 | 0.950 | 0.986 | 0.997 | 1.000 | 1.000 | 1.000 | 1.000 | 1.000 | 1.000 | 1.000 | 1.000 | 1.000 |
| 45 | 15 | 0.000 | 0.000 | 0.000 | 0.000 | 0.009 | 0.052 | 0.160 | 0.362 | 0.592 | 0.898 | 0.986 | 0.999 | 1.000 | 1.000 | 1.000 | 1.000 | 1.000 | 1.000 |
| 60 | 15 | 0.000 | 0.000 | 0.000 | 0.000 | 0.000 | 0.000 | 0.000 | 0.003 | 0.018 | 0.186 | 0.571 | 0.866 | 0.974 | 0.997 | 1.000 | 1.000 | 1.000 | 1.000 |
| 75 | 15 | 0.000 | 0.000 | 0.000 | 0.000 | 0.000 | 0.000 | 0.000 | 0.000 | 0.000 | 0.001 | 0.028 | 0.226 | 0.578 | 0.850 | 0.964 | 0.994 | 0.999 | 1.000 |
| 90 | 15 | 0.000 | 0.000 | 0.000 | 0.000 | 0.000 | 0.000 | 0.000 | 0.000 | 0.000 | 0.000 | 0.000 | 0.003 | 0.053 | 0.269 | 0.598 | 0.846 | 0.958 | 0.991 |
| 20 | 20 | 0.714 | 0.908 | 0.979 | 0.997 | 1.000 | 1.000 | 1.000 | 1.000 | 1.000 | 1.000 | 1.000 | 1.000 | 1.000 | 1.000 | 1.000 | 1.000 | 1.000 | 1.000 |
| 40 | 20 | 0.000 | 0.011 | 0.066 | 0.197 | 0.448 | 0.697 | 0.864 | 0.950 | 0.985 | 0.999 | 1.000 | 1.000 | 1.000 | 1.000 | 1.000 | 1.000 | 1.000 | 1.000 |
| 60 | 20 | 0.000 | 0.000 | 0.000 | 0.000 | 0.001 | 0.012 | 0.047 | 0.172 | 0.373 | 0.770 | 0.953 | 0.994 | 1.000 | 1.000 | 1.000 | 1.000 | 1.000 | 1.000 |
| 80 | 20 | 0.000 | 0.000 | 0.000 | 0.000 | 0.000 | 0.000 | 0.000 | 0.000 | 0.003 | 0.066 | 0.351 | 0.718 | 0.923 | 0.986 | 0.998 | 1.000 | 1.000 | 1.000 |
| 100 | 20 | 0.000 | 0.000 | 0.000 | 0.000 | 0.000 | 0.000 | 0.000 | 0.000 | 0.000 | 0.000 | 0.004 | 0.089 | 0.358 | 0.695 | 0.901 | 0.977 | 0.996 | 0.999 |
| 25 | 25 | 0.558 | 0.823 | 0.949 | 0.989 | 0.998 | 1.000 | 1.000 | 1.000 | 1.000 | 1.000 | 1.000 | 1.000 | 1.000 | 1.000 | 1.000 | 1.000 | 1.000 | 1.000 |
| 50 | 25 | 0.000 | 0.003 | 0.024 | 0.087 | 0.290 | 0.550 | 0.763 | 0.898 | 0.963 | 0.997 | 1.000 | 1.000 | 1.000 | 1.000 | 1.000 | 1.000 | 1.000 | 1.000 |
| 30 | 30 | 0.416 | 0.728 | 0.909 | 0.977 | 0.995 | 0.999 | 1.000 | 1.000 | 1.000 | 1.000 | 1.000 | 1.000 | 1.000 | 1.000 | 1.000 | 1.000 | 1.000 | 1.000 |
| 60 | 30 | 0.000 | 0.001 | 0.008 | 0.036 | 0.189 | 0.429 | 0.662 | 0.837 | 0.934 | 0.993 | 1.000 | 1.000 | 1.000 | 1.000 | 1.000 | 1.000 | 1.000 | 1.000 |
| 40 | 40 | 0.209 | 0.553 | 0.816 | 0.941 | 0.985 | 0.997 | 0.999 | 1.000 | 1.000 | 1.000 | 1.000 | 1.000 | 1.000 | 1.000 | 1.000 | 1.000 | 1.000 | 1.000 |
| 80 | 40 | 0.000 | 0.000 | 0.001 | 0.005 | 0.086 | 0.260 | 0.479 | 0.705 | 0.862 | 0.981 | 0.998 | 1.000 | 1.000 | 1.000 | 1.000 | 1.000 | 1.000 | 1.000 |
| 50 | 50 | 0.096 | 0.417 | 0.723 | 0.896 | 0.970 | 0.993 | 0.999 | 1.000 | 1.000 | 1.000 | 1.000 | 1.000 | 1.000 | 1.000 | 1.000 | 1.000 | 1.000 | 1.000 |
| 100 | 50 | 0.000 | 0.000 | 0.000 | 0.001 | 0.042 | 0.156 | 0.332 | 0.577 | 0.780 | 0.962 | 0.996 | 1.000 | 1.000 | 1.000 | 1.000 | 1.000 | 1.000 | 1.000 |

**Table 6.** Distribution of the multinomial minimum. For example, in the case with $n = 25$ and $m = 5$, $P(\min N_i \leq 2; n = 25, m = 5) = 0.446$.

| | | t | | | | | | | |
|---|---|---|---|---|---|---|---|---|---|
| n | m | 0 | 1 | 2 | 3 | 4 | 5 | 6 | 7 |
| 5 | 5 | 0.962 | 1.000 | 1.000 | 1.000 | 1.000 | 1.000 | 1.000 | 1.000 |
| 10 | 5 | 0.477 | 0.988 | 1.000 | 1.000 | 1.000 | 1.000 | 1.000 | 1.000 |
| 15 | 5 | 0.171 | 0.680 | 0.994 | 1.000 | 1.000 | 1.000 | 1.000 | 1.000 |
| 20 | 5 | 0.057 | 0.325 | 0.783 | 0.997 | 1.000 | 1.000 | 1.000 | 1.000 |
| 25 | 5 | 0.019 | 0.135 | 0.446 | 0.844 | 0.998 | 1.000 | 1.000 | 1.000 |
| 30 | 5 | 0.006 | 0.052 | 0.214 | 0.539 | 0.882 | 0.999 | 1.000 | 1.000 |
| 40 | 5 | 0.001 | 0.007 | 0.040 | 0.140 | 0.356 | 0.669 | 0.926 | 0.999 |
| 30 | 10 | 0.371 | 0.931 | 0.999 | 1.000 | 1.000 | 1.000 | 1.000 | 1.000 |
| 40 | 10 | 0.142 | 0.614 | 0.972 | 0.999 | 1.000 | 1.000 | 1.000 | 1.000 |
| 50 | 10 | 0.051 | 0.306 | 0.757 | 0.986 | 1.000 | 1.000 | 1.000 | 1.000 |
| 60 | 10 | 0.018 | 0.133 | 0.450 | 0.841 | 0.993 | 1.000 | 1.000 | 1.000 |
| 80 | 10 | 0.003 | 0.022 | 0.104 | 0.319 | 0.657 | 0.926 | 0.997 | 0.999 |
| 90 | 15 | 0.03 | 0.208 | 0.616 | 0.945 | 0.999 | 1.000 | 1.000 | 1.000 |
| 100 | 20 | 0.113 | 0.556 | 0.954 | 0.999 | 1.000 | 1.000 | 1.000 | 1.000 |

**Table 7.** Distribution of the multinomial range. For example, in the case of $n = 15$ and $m = 15$, $P(\max N_i - \min N_i \leq 3; n = 15, m = 15) = 0.51$.

| | | t | | | | | | | | | | | | | | | |
|---|---|---|---|---|---|---|---|---|---|---|---|---|---|---|---|---|---|
| n | m | 1 | 2 | 3 | 4 | 5 | 6 | 7 | 8 | 9 | 10 | 11 | 12 | 13 | 14 | 15 | 16 |
| 5 | 5 | 0.038 | 0.710 | 0.966 | 0.998 | 1.000 | 1.000 | 1.000 | 1.000 | 1.000 | 1.000 | 1.000 | 1.000 | 1.000 | 1.000 | 1.000 | 1.000 |
| 10 | 5 | 0.012 | 0.321 | 0.601 | 0.867 | 0.971 | 0.996 | 1.000 | 1.000 | 1.000 | 1.000 | 1.000 | 1.000 | 1.000 | 1.000 | 1.000 | 1.000 |
| 15 | 5 | 0.006 | 0.181 | 0.386 | 0.659 | 0.854 | 0.953 | 0.988 | 0.997 | 1.000 | 1.000 | 1.000 | 1.000 | 1.000 | 1.000 | 1.000 | 1.000 |
| 20 | 5 | 0.003 | 0.116 | 0.265 | 0.500 | 0.716 | 0.868 | 0.949 | 0.983 | 0.995 | 0.999 | 1.000 | 1.000 | 1.000 | 1.000 | 1.000 | 1.000 |
| 25 | 5 | 0.002 | 0.081 | 0.193 | 0.388 | 0.595 | 0.770 | 0.887 | 0.952 | 0.982 | 0.994 | 0.998 | 0.998 | 0.999 | 1.000 | 1.000 | 1.000 |
| 30 | 5 | 0.001 | 0.059 | 0.146 | 0.308 | 0.497 | 0.676 | 0.815 | 0.906 | 0.958 | 0.983 | 0.994 | 0.998 | 0.999 | 1.000 | 1.000 | 1.000 |
| 10 | 10 | 0.000 | 0.396 | 0.873 | 0.984 | 0.999 | 1.000 | 1.000 | 1.000 | 1.000 | 1.000 | 1.000 | 1.000 | 1.000 | 1.000 | 1.000 | 1.000 |
| 20 | 10 | 0.000 | 0.060 | 0.237 | 0.640 | 0.896 | 0.977 | 0.996 | 0.999 | 1.000 | 1.000 | 1.000 | 1.000 | 1.000 | 1.000 | 1.000 | 1.000 |
| 30 | 10 | 0.000 | 0.016 | 0.079 | 0.313 | 0.612 | 0.841 | 0.949 | 0.986 | 0.997 | 0.999 | 1.000 | 1.000 | 1.000 | 1.000 | 1.000 | 1.000 |
| 40 | 10 | 0.000 | 0.006 | 0.032 | 0.158 | 0.380 | 0.637 | 0.831 | 0.935 | 0.978 | 0.994 | 0.998 | 1.000 | 1.000 | 1.000 | 1.000 | 1.000 |
| 50 | 10 | 0.000 | 0.002 | 0.015 | 0.086 | 0.236 | 0.458 | 0.679 | 0.840 | 0.932 | 0.974 | 0.991 | 0.997 | 0.999 | 1.000 | 1.000 | 1.000 |
| 60 | 10 | 0.000 | 0.001 | 0.008 | 0.050 | 0.151 | 0.325 | 0.535 | 0.724 | 0.858 | 0.935 | 0.973 | 0.990 | 0.997 | 0.999 | 1.000 | 1.000 |
| 15 | 15 | 0.000 | 0.061 | 0.510 | 0.875 | 0.978 | 0.997 | 1.000 | 1.000 | 1.000 | 1.000 | 1.000 | 1.000 | 1.000 | 1.000 | 1.000 | 1.000 |
| 30 | 15 | 0.010 | 0.016 | 0.079 | 0.313 | 0.612 | 0.841 | 0.949 | 0.986 | 0.997 | 0.999 | 1.000 | 1.000 | 1.000 | 1.000 | 1.000 | 1.000 |
| 45 | 15 | 0.000 | 0.001 | 0.028 | 0.111 | 0.299 | 0.542 | 0.756 | 0.892 | 0.959 | 0.986 | 0.996 | 0.999 | 1.000 | 1.000 | 1.000 | 1.000 |
| 60 | 15 | 0.000 | 0.001 | 0.008 | 0.050 | 0.151 | 0.325 | 0.535 | 0.724 | 0.858 | 0.935 | 0.973 | 0.990 | 0.997 | 0.999 | 1.000 | 1.000 |
| 75 | 15 | 0.000 | 0.000 | 0.005 | 0.023 | 0.082 | 0.198 | 0.365 | 0.553 | 0.719 | 0.843 | 0.920 | 0.963 | 0.984 | 0.994 | 0.998 | 0.999 |
| 90 | 15 | 0.000 | 0.000 | 0.002 | 0.013 | 0.047 | 0.124 | 0.250 | 0.412 | 0.581 | 0.729 | 0.841 | 0.914 | 0.957 | 0.980 | 0.991 | 0.996 |
| 20 | 20 | 0.000 | 0.121 | 0.706 | 0.949 | 0.993 | 0.999 | 1.000 | 1.000 | 1.000 | 1.000 | 1.000 | 1.000 | 1.000 | 1.000 | 1.000 | 1.000 |
| 40 | 20 | 0.000 | 0.002 | 0.029 | 0.321 | 0.745 | 0.934 | 0.986 | 0.997 | 1.000 | 1.000 | 1.000 | 1.000 | 1.000 | 1.000 | 1.000 | 1.000 |
| 60 | 20 | 0.000 | 0.000 | 0.002 | 0.065 | 0.274 | 0.607 | 0.850 | 0.954 | 0.988 | 0.997 | 0.999 | 1.000 | 1.000 | 1.000 | 1.000 | 1.000 |
| 80 | 20 | 0.000 | 0.000 | 0.000 | 0.015 | 0.089 | 0.301 | 0.586 | 0.812 | 0.929 | 0.977 | 0.993 | 0.998 | 1.000 | 1.000 | 1.000 | 1.000 |
| 100 | 20 | 0.000 | 0.000 | 0.000 | 0.004 | 0.030 | 0.138 | 0.349 | 0.603 | 0.802 | 0.917 | 0.969 | 0.990 | 0.997 | 0.999 | 1.000 | 1.000 |
| 25 | 25 | 0.000 | 0.067 | 0.634 | 0.931 | 0.991 | 0.999 | 1.000 | 1.000 | 1.000 | 1.000 | 1.000 | 1.000 | 1.000 | 1.000 | 1.000 | 1.000 |
| 50 | 25 | 0.000 | 0.000 | 0.010 | 0.227 | 0.680 | 0.912 | 0.981 | 0.996 | 0.999 | 1.000 | 1.000 | 1.000 | 1.000 | 1.000 | 1.000 | 1.000 |

(Continued.)

**Table 7.** (*Continued.*)

| n | m | 1 | 2 | 3 | 4 | 5 | 6 | 7 | 8 | 9 | 10 | 11 | 12 | 13 | 14 | 15 | 16 |
|---|---|---|---|---|---|---|---|---|---|---|---|---|---|---|---|---|---|
| 75 | 25 | 0.000 | 0.000 | 0.000 | 0.030 | 0.178 | 0.505 | 0.799 | 0.936 | 0.983 | 0.996 | 0.999 | 1.000 | 1.000 | 1.000 | 1.000 | 1.000 |
| 100 | 25 | 0.000 | 0.000 | 0.000 | 0.005 | 0.041 | 0.201 | 0.483 | 0.747 | 0.901 | 0.967 | 0.990 | 0.997 | 0.999 | 1.000 | 1.000 | 1.000 |
| 30 | 30 | 0.000 | 0.037 | 0.570 | 0.914 | 0.988 | 0.999 | 1.000 | 1.000 | 1.000 | 1.000 | 1.000 | 1.000 | 1.000 | 1.000 | 1.000 | 1.000 |
| 60 | 30 | 0.000 | 0.000 | 0.003 | 0.161 | 0.620 | 0.891 | 0.975 | 0.995 | 0.999 | 1.000 | 1.000 | 1.000 | 1.000 | 1.000 | 1.000 | 1.000 |
| 40 | 40 | 0.000 | 0.000 | 0.201 | 0.720 | 0.941 | 0.990 | 0.999 | 1.000 | 1.000 | 1.000 | 1.000 | 1.000 | 1.000 | 1.000 | 1.000 | 1.000 |
| 80 | 40 | 0.000 | 0.000 | 0.001 | 0.027 | 0.195 | 0.587 | 0.857 | 0.960 | 0.990 | 0.998 | 1.000 | 1.000 | 1.000 | 1.000 | 1.000 | 1.000 |
| 50 | 50 | 0.000 | 0.003 | 0.370 | 0.848 | 0.976 | 0.997 | 1.000 | 1.000 | 1.000 | 1.000 | 1.000 | 1.000 | 1.000 | 1.000 | 1.000 | 1.000 |
| 100 | 50 | 0.000 | 0.000 | 0.000 | 0.042 | 0.430 | 0.811 | 0.954 | 0.991 | 0.998 | 1.000 | 1.000 | 1.000 | 1.000 | 1.000 | 1.000 | 1.000 |

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
