## [Reviewer comments · Royal Society Open Science]

Review History

RSOS-190198.R0 (Original submission)

Review form: Reviewer 1

Is the manuscript scientifically sound in its present form?

Yes

Are the interpretations and conclusions justified by the results?

Yes

Is the language acceptable?

Yes

Is it clear how to access all supporting data?

Yes

Do you have any ethical concerns with this paper?

No

Have you any concerns about statistical analyses in this paper?

Yes

Recommendation?

Major revision is needed (please make suggestions in comments)

Comments to the Author(s)

See report (Appendix A).

Review form: Reviewer 2

Is the manuscript scientifically sound in its present form?

Yes

Are the interpretations and conclusions justified by the results?

Yes

Is the language acceptable?

Yes

Is it clear how to access all supporting data?

Yes

Do you have any ethical concerns with this paper?

No

Have you any concerns about statistical analyses in this paper?

No

Recommendation?

Accept with minor revision (please list in comments)

Comments to the Author(s)

Please see the attached file (Appendix B).

Decision letter (RSOS-190198.R0)

10-Jun-2019

Dear Professor Cirillo

On behalf of the Editors, I am pleased to inform you that your Manuscript RSOS-190198 entitled "Computing the Exact Distributions of Some Functions of the Ordered Multinomial Counts:

Maximum, Minimum, Range and Sums of Order Statistics" has been accepted for publication in Royal Society Open Science subject to minor revision in accordance with the referee suggestions. Please find the referees' comments at the end of this email.

The reviewers and handling editors have recommended publication, but also suggest some minor revisions to your manuscript. Therefore, I invite you to respond to the comments and revise your manuscript.

- Ethics statement

- Data accessibility

If you wish to submit your supporting data or code to Dryad (<http://datadryad.org/>), or modify your current submission to dryad, please use the following link:
<http://datadryad.org/submit?journalID=RSOS&manu=RSOS-190198>

- Competing interests

- Authors' contributions

- Acknowledgements

- Funding statement

Because the schedule for publication is very tight, it is a condition of publication that you submit the revised version of your manuscript before 19-Jun-2019. Please note that the revision deadline will expire at 00.00am on this date. If you do not think you will be able to meet this date please let me know immediately.

Supplementary files will be published alongside the paper on the journal website and posted on the online figshare repository (<https://rs.figshare.com/>). The heading and legend provided for each supplementary file during the submission process will be used to create the figshare page,

so please ensure these are accurate and informative so that your files can be found in searches. Files on figshare will be made available approximately one week before the accompanying article so that the supplementary material can be attributed a unique DOI.

on behalf of Professor Andreas Kyprianou (Associate Editor) and Mark Chaplain (Subject Editor)
openscience@royalsociety.org

Associate Editor Comments to Author (Professor Andreas Kyprianou):

We have two good reviews. There is lots to do to clean up and some suggestions to follow, which I would strongly encourage you to think about, but nothing that seems to be critical.

Reviewer comments to Author:
Reviewer: 1

Comments to the Author(s)
See report

Reviewer: 2

Comments to the Author(s)
Please see the attached file.

Author's Response to Decision Letter for (RSOS-190198.R0)

See Appendix C.

RSOS-190198.R1 (Revision)

Review form: Reviewer 1

Is the manuscript scientifically sound in its present form?

Yes

Are the interpretations and conclusions justified by the results?

Yes

Is the language acceptable?

Yes

Do you have any ethical concerns with this paper?

No

Have you any concerns about statistical analyses in this paper?

No

Recommendation?

Accept as is

Comments to the Author(s)

I am happy with the authors' responses to my comments.

Decision letter (RSOS-190198.R1)

19-Aug-2019

Dear Professor Cirillo,

I am pleased to inform you that your manuscript entitled "Computing the Exact Distributions of Some Functions of the Ordered Multinomial Counts: Maximum, Minimum, Range and Sums of Order Statistics" is now accepted for publication in Royal Society Open Science.

Royal Society Open Science operates under a continuous publication model (<http://bit.ly/cpFAQ>). Your article will be published straight into the next open issue and this will be the final version of the paper. As such, it can be cited immediately by other researchers.

As the issue version of your paper will be the only version to be published I would advise you to check your proofs thoroughly as changes cannot be made once the paper is published.

on behalf of Professor Andreas Kyprianou (Associate Editor) and Mark Chaplain (Subject Editor)
openscience@royalsociety.org

Reviewer comments to Author:
Reviewer: 1

Comments to the Author(s)
I am happy with the authors' responses to my comments.

Appendix A

Referee report on:

Computing the Exact Distributions of Some Functions of the Ordered Multinomial Counts: Maximum, Minimum, Range and Sums of Order Statistics

By: Bonetti, Cirillo and Ogay

The multinomial distribution indeed plays a very important role in modern statistics. The manuscript provides algorithms for deriving the exact distribution of sums and ranges of the largest multinomial counts order statistics.

I think it is an interesting and very nicely written paper. I really like the idea of replacing chi-squared distributed goodness of fit tests with more powerful tests for which it is more difficult to compute p-values. I also think that the authors do a very nice job presenting the algorithm and motivating the use of these statistics. My main concern is that explicitly deriving the test statistic distributions may not be needed in practice. For example, suppose the goal of the analysis is to test the null hypothesis of equal multinomial probabilities at level $\alpha = 0.05$ using the maximum multinomial count statistic. One option would be to use the author's code to derive the statistic's null distribution; compute the p-value by summing the null probability for the statistic value that are greater than or equal to the statistic value for the observed data; reject the null hypothesis if the p-value is less than 0.05. I think an easier option would be a Monte Carlo simulation that takes several minutes to write and several seconds to run: sample 10^6 iid null Multinomial vectors; compute the proportion of samples for which the statistic value is greater than or equal to the statistic value for the observed data; reject the null hypothesis if the proportion is less than 0.05. Here is R code that runs in 3 seconds that evaluates the exact distribution of the maximum:

```
> date()
[1] "Tue May 28 14:18:50 2019"
> aa<- rmultinom(10^6,size = 100,prob = rep(1/30,30))
> table(apply(aa,2,max)) / 10^6
 5      6      7      8      9     10     11     12     13     14     15     16     17
0.004937 0.154291 0.377932 0.283428 0.122928 0.040990 0.011638 0.002994 0.000701 0.000134 0.000022 0.000004 0.000001

> date()
[1] "Tue May 28 14:18:53 2019"
```

I also found the comparison between approximation and exact results in Section 7 a little misleading. I think that presenting figures comparing the power between 0.05 significance level tests specified by either approximations or exact results may mislead the reader to think that the test based on the exact computations is more powerful than the test based on the approximation when in facts the two tests use the same test statistic with different (random) cutoffs. For the exact computation the test's type I error probability is

exactly 0.05 and for the approximation it is smaller than 0.05. However in examples in which the approximation underestimates the statistic's tail probabilities the test based on the approximation would have type I error greater than 0.05 and would be more powerful than the exact test. Furthermore, the power comparison highlights the fact that to compute power (i.e. compute test statistic tail probabilities for the non-equiprobable case) you need to resort to MC simulation. In general, for presenting the results of MC probability assessment I suggest adding standard errors. Looking at Figures 4 and 5, I also think the decrease in power of the of exact test when increasing N is bad advertisement for use of the statistic in this particular example. Why not simply show how bad the existing approximations evaluate the test statistics tail probabilities?

Appendix B

Comments on

„Computing the Exact Distributions of Some Functions of the Ordered Multinomial Counts: Maximum, Minimum, Range and Sums of Order Statistics“

The authors derive the exact distributions of the maximum, the minimum, the range, and the sum of the J largest order statistics of a random vector having an equiprobable multinomial distribution based upon an unpublished work by Michael Arnold Rapoport. Preparing these main results, the authors discuss some approximations and some exact results of the distribution of the maximum, the minimum and the range, respectively. Afterwards, their exact results are compared to those approximations and applied in statistical testing theory. Finally, the authors illustrate two applications to testing for the homogeneity of a Poisson process and for clustering diseases.

Overall, the manuscript is well structured and well written. After some minor concerns listed below, I recommend accepting this paper for publication.

My minor concerns are:

- 1) page 4, line 9: “Dasgupta [13]”
- 2) page 4, equation (3.1): Please reformulate to get equation (3.1) in one line.
For instance: “The transition probability $P(S_k = s_k | S_{k-1} = s_{k-1}; p_k^*)$ from $S_{k-1} = s_{k-1}$ to $S_k = s_k$ is denoted by $P(s_k | s_{k-1}; p_k^*)$ for brevity, where $p_k^* = \dots$. It is
$$P(s_k | s_{k-1}; p_k^*) = \dots \quad (3.1)$$
- 3) Drop “=” at the end of the first line if an equation expands over two lines.
For example: page 5, lines 44-48 or page 7, lines 19-23
- 4) page 7, line 8: Put “r-1” in math mode
- 5) Choose tall bracket [...] in equation (4.2) and the following ones.
- 6) Control the line breaks.
For example: page 7, lines 40-42 or page 8, lines 41-44
- 7) page 11, lines 21-24: Correct the representation of the equation.
- 8) page 12: The abbreviations “cdf” in Figure 2 and “i.i.d.” in line 27 are not introduced.
- 9) page 13, lines 52/53: “ $LN(\mu_{LN}, \sigma_{LN})$ with $\mu_{LN} = 0$ and”
- 10) Put the figure closer to the corresponding text.
For example Figures 2 and 3 are illustrated at page 12, but textually mentioned just at page 13.
- 11) page 19, line 39: “from”
- 12) The readability of the algorithms given in the Appendix and the comparability to the results could be improved by using the notations from the corresponding section.
For example use n and m instead of N and I , respectively.

Appendix C

Dear Referees,

First of all thanks for your nice comments on our work and for the useful suggestions you gave us.

As you will see, we have accepted all the changes you have proposed and commented upon the points you have raised.

Here below we collect all the changes introduced in the paper.

Thanks for your help and best regards.

The Authors.

Reviewer 1

- *“My main concern is that explicitly deriving the test statistic distributions may not be needed in practice...”*

We agree on the fact that, nowadays, very good approximations for the distributions of functions of the multinomial counts can be obtained via simulations, as suggested in your report. And probably, for most applications, those approximations could be sufficient. However, from an epistemological point of view, they will always be approximations, and not exact results as those we propose. We feel that our paper contributes to the resolution of an old open problem related to multinomial random variables.

To take into account your comment, we have added the following text just before Section 2(a):

Note that, given the increased (and still increasing) computing power one can rely upon today, the probability distributions of functions of the multinomial counts can also be estimated via Monte Carlo simulations. However, even if extremely accurate, from a conceptual point of view they are still approximations and not exact results, as those we will discuss in this article.

- *“In general, for presenting the results of MC probability assessment I suggest adding standard errors...”*

Regarding your comment about standard errors in the MC simulation, we have added the following text on Page 13, Paragraph 3 (the formula is in Latex):

For both figures, standard errors can be computed using the simple formula $\sqrt{\frac{p(1-p)}{3000}}$, where p is the estimated power.

This gives the reader a quick way to compute the standard errors at her own discretion.

- *“I also found the comparison between approximation and exact results in Section 7 a little misleading...”*

Finally, to address your very relevant point about power comparisons, we have clarified the text, by adding the following lines at the end of Section 7:

However, one should keep in mind that the observed differences in the performances of the two procedures (exact vs approximation) may also depend upon the accuracy of the calculations of the tail probabilities under the approximation formulas, which may produce type I error probabilities different from the desired ones.

Thanks a lot for your help in clarifying our paper.

Reviewer 2

Following your suggestions, we have modified the paper as follows.

- 1) page 4, line 9: "Dasgupta [13]"
DONE (also on page 3)
- 2) page 4, equation (3.1): Please reformulate to get equation (3.1) in one line.
For instance: "The transition probability $P(S_k = sk | S_{k-1} = sk-1; pk^*)$ from $S_{k-1} = sk-1$ to $S_k = sk$ is denoted by $P(sk|sk-1;pk^*)$ for brevity, where $pk^* = \dots$. It is $P(sk|sk-1;pk^*) = \dots$ (3.1)
DONE
- 3) Drop "=" at the end of the first line if an equation expands over two lines.
For example: page 5, lines 44-48 or page 7, lines 19-23
DONE for all equations
- 4) page 7, line 8: Put "r-1" in math mode
DONE
- 5) Choose tall bracket $\lfloor \dots \rfloor$ in equation (4.2) and the following ones.
DONE
- 6) Control the line breaks. For example: page 7, lines 40-42 or page 8, lines 41-44
DONE, in line with the style requirements of the journal.
- 7) page 11, lines 21-24: Correct the representation of the equation.
DONE
- 8) page 12: The abbreviations "cdf" in Figure 2 and "i.i.d." in line 27 are not introduced.
DONE in the figure and in the text.
- 9) page 13, lines 52/53: " $LN(\mu_{LN}, \sigma_{LN})$ with $\mu_{LN} = 0$ and"
DONE
- 10) Put the figure closer to the corresponding text.
For example Figures 2 and 3 are illustrated at page 12, but textually mentioned just at page 13.
DONE
- 11) page 19, line 39: "from"
DONE
- 12) The readability of the algorithms given in the Appendix and the comparability to the results could be improved by using the notations from the corresponding section. For example use n and m instead of N and I , respectively.
DONE

Thanks a lot for your help in finding even the smallest typo. We appreciate that.